# A Short Survey on Importance Weighting for Machine Learning

**Masanari Kimura**
*School of Mathematics and Statistics*
*The University of Melbourne*

*m.kimura@unimelb.edu.au*

**Hideitsu Hino**
*The Institute of Statistical Mathematics*
*Center for Advanced Intelligence Project, RIKEN*

*hino@ism.ac.jp*

## Abstract

Importance weighting is a fundamental procedure in statistics and machine learning that weights the objective function or probability distribution based on the importance of the instance in some sense. The simplicity and usefulness of the idea has led to many applications of importance weighting. For example, it is known that supervised learning under an assumption about the difference between the training and test distributions, called distribution shift, can guarantee statistically desirable properties through importance weighting by their density ratio. This survey summarizes the broad applications of importance weighting in machine learning and related research.

## 1 Introduction

Taking population or empirical average of a given set of instances is an essential procedure in machine learning and statistical data analysis. In this case, it is frequent that each instance plays a different role in terms of importance. For example, consider the expectation

$$I = \mathbb{E}_{\pi(\boldsymbol{x})}\left[g(\boldsymbol{x})\right] = \int_{\mathcal{X}} g(\boldsymbol{x})\pi(\boldsymbol{x})d\boldsymbol{x} \tag{1}$$

of any function $g(\boldsymbol{x})$ of a random variable $\boldsymbol{x} \sim \pi(\boldsymbol{x})$. If we denote the independent samples from $\pi(\boldsymbol{x})$ by $\{\boldsymbol{x}_1, \ldots, \boldsymbol{x}_T\}$, the expected value to be obtained can be estimated as follows:

$$\hat{I} = \frac{1}{T}\sum_{t=1}^{T} g(\boldsymbol{x}_t). \tag{2}$$

From the law of large numbers, $\hat{I}$ converges to $I$ when $T \to \infty$, and such an approximation of the expectation is called a Monte Carlo integration. For applications to Bayesian statistics, $\pi(\boldsymbol{x})$ corresponds to the posterior distribution. In many cases, the posterior distribution is so complex that it is difficult to sample directly from $\pi(\boldsymbol{x})$. Then, considering another probability distribution $q(\boldsymbol{x})$, which allows for easy sampling, we have

$$I = \int_{\mathcal{X}} g(\boldsymbol{x})\pi(\boldsymbol{x})d\boldsymbol{x} = \int_{\mathcal{X}} g(\boldsymbol{x})\frac{\pi(\boldsymbol{x})}{q(\boldsymbol{x})}q(\boldsymbol{x})d\boldsymbol{x} = \mathbb{E}_{q(\boldsymbol{x})}\left[g(\boldsymbol{x})\frac{\pi(\boldsymbol{x})}{q(\boldsymbol{x})}\right].$$

Then, by using $w(\boldsymbol{x}) = \pi(\boldsymbol{x})/q(\boldsymbol{x})$, we can estimate $I$ as

$$\hat{I} = \frac{1}{T}\sum_{t=1}^{T} w(\boldsymbol{x}_t)g(\boldsymbol{x}_t). \tag{3}$$

**Distribution Shift Adaptation (Sec. 3)**

**Importance Weighted ERM**

- Covariate Shift (Sec. 3.1)
- Target Shift (Sec. 3.2)
- Sample Selection Bias (Sec. 3.3)
- Subpopulation Shift (Sec. 3.4)
- Feedback Shift (Sec. 3.5)
- Domain Adaptation (Sec. 3.6)
  - Multi-Source Domain Adaptation (Sec. 3.6.1)
  - Partial Domain Adaptation (Sec. 3.6.2)
  - Open-Set Domain Adaptation (Sec. 3.6.3)
  - Universal Domain Adaptation (Sec. 3.6.4)

**Distributionally Robust Optimization (Sec. 3.7)**
Constructing uncertainty set

**Active Learning (Sec. 4)**
Sample selection for labeling

**Model Calibration (Sec. 5)**
Down-weighting well-classified instances

**PU Learning (Sec. 6)**
Weighting unlabeled data according to positivity

**Label Noise Correction (Sec. 7)**
Weighting as the label noise probability

**Subgroup Fairness (Sec. 8)**
Weighting to groups based on sensitive features.

Figure 1: Applications of importance weighting.

This technique is known as importance sampling (Kahn & Harris, 1951; Tokdar & Kass, 2010; Owen & Zhou, 2000) and is a useful procedure in computational statistics.

This simple and effective idea has been adapted in various areas of machine learning in recent years. A similar procedure is called importance weighting in many machine learning studies, and its usefulness in various applied tasks has been reported. Figure 1 summarizes the subfields of machine learning for which importance weighting has been reported to be useful. This survey focuses on summarizing examples where importance weighting is used in machine learning.

## 2 Preliminary

First, we formulate the problem of supervised learning. We denote by $\mathcal{X} \subset \mathbb{R}^d$ the $d$-dimensional input space, where $d \in \mathbb{N}$. The output space is denoted by $\mathcal{Y} \subset \mathbb{R}$ (regression) or $\mathcal{Y} \subset \{1, \ldots, K\}$ ($K$-class classification). We assume that training examples $\{(\boldsymbol{x}_i^{tr}, y_i^{tr})\}_{i=1}^{n_{tr}}$ are independently and identically distributed (i.i.d.) according to some fixed but unknown distribution $p_{tr}(\boldsymbol{x}, y)$, which can be decomposed into the marginal distribution and the conditional probability distribution, i.e., $p_{tr}(\boldsymbol{x}, y) = p_{tr}(\boldsymbol{x})p_{tr}(y|\boldsymbol{x})$. We also denote the test examples by $\{(\boldsymbol{x}_i^{te}, y_i^{te})\}_{i=1}^{n_{te}}$ drawn from a test distribution $p_{te}(\boldsymbol{x}, y) = p_{te}(\boldsymbol{x})p_{te}(y|\boldsymbol{x})$. Here $n_{tr} \in \mathbb{N}$ and $n_{te} \in \mathbb{N}$ are training and test sample size, respectively. Let $\mathcal{H}$ be a hypothesis class. The goal of supervised learning is to obtain a hypothesis $h : \mathcal{X} \to \mathbb{R}$ ($h \in \mathcal{H}$) with the training examples that minimizes the expected loss over the test distribution.

**Definition 1** (Expected error). Given a hypothesis $h \in \mathcal{H}$ and probability distribution $p(\boldsymbol{x}, y)$ the expected error $\mathcal{R}(h; p)$ is defined as

$$\mathcal{R}(h; p) \coloneqq \mathbb{E}_{(\boldsymbol{x}, y) \sim p(\boldsymbol{x}, y)}\Big[\ell(h(\boldsymbol{x}), y)\Big], \tag{4}$$

where $\ell : \mathbb{R} \times \mathcal{Y} \to \mathbb{R}$ is the loss function that measures the discrepancy between the true output value $y$ and the predicted value $\hat{y} \coloneqq h(\boldsymbol{x})$.

In this work, we assume that $\ell$ is bounded from above, i.e., for some constant $C > 0$, $\ell(y, y') < C$ ($\forall y, y' \in \mathcal{Y}$). The expected error of a hypothesis $h$ is not directly accessible since the underlying distribution $p(\boldsymbol{x}, y)$ is unknown. Then, we have to measure the empirical error of hypothesis $h$ on the observable labeled sample.

**Definition 2** (Empirical error). Given a hypothesis $h \in \mathcal{H}$ and labeled sample $S_n = \{(\boldsymbol{x}_i, y_i)\}_{i=1}^n$, the empirical error $\hat{\mathcal{R}}(h; S_n)$ is defined as

$$\hat{\mathcal{R}}(h; S_n) := \frac{1}{n} \sum_{i=1}^n \ell(h(\boldsymbol{x}_i), y_i), \tag{5}$$

where $\ell : \mathbb{R} \times \mathcal{Y} \to \mathbb{R}$ is the loss function.

The procedure of minimizing Eq. 5, called Empirical Risk Minimization (ERM), is known to be useful in learning with i.i.d. settings (Sugiyama, 2015; James et al., 2013; Hastie et al., 2009; Bishop & Nasrabadi, 2006; Vapnik, 1999a;b). For example, we assume that the labeled sample $S$ is a set of i.i.d. examples according to distribution $p(\boldsymbol{x}, y)$. Then, the empirical error $\hat{\mathcal{R}}(h; S_n)$ is an unbiased estimator of the expected error $\mathcal{R}(h; p)$ as follows.

$$\mathbb{E}_{p(\boldsymbol{x}, y)}\left[\hat{\mathcal{R}}(h; S_n)\right] = \frac{1}{n} \sum_{i=1}^n \mathbb{E}_{p(\boldsymbol{x}, y)}\left[\ell(h(\boldsymbol{x}), y)\right] = \frac{1}{n} \sum_{i=1}^n \mathcal{R}(h; p) = \mathcal{R}(h; p). \tag{6}$$

## 2.1 Density Ratio Estimation

Density ratio estimation is a method in machine learning used to discern the difference between the distributions of two data sets. Its objective is to calculate the density ratio, denoted as $r(\boldsymbol{x}) = p_{te}(\boldsymbol{x})/p_{tr}(\boldsymbol{x})$, which expresses the ratio of the probability density functions between the test distribution $p_{te}(\boldsymbol{x})$ and the training distribution $p_{tr}(\boldsymbol{x})$.

Moment matching is one of the most prevalent frameworks in density ratio estimation. This approach focuses on aligning the moments of two distributions, $\hat{p}_{te}(\boldsymbol{x}) = \hat{r}(\boldsymbol{x})p_{tr}(\boldsymbol{x})$ and $p_{te}(\boldsymbol{x})$. One common example is matching their means, as shown by the equation:

$$\int \boldsymbol{x} \cdot \hat{r}(\boldsymbol{x})p_{tr}(\boldsymbol{x})d\boldsymbol{x} = \int \boldsymbol{x} \cdot p_{te}(\boldsymbol{x})d\boldsymbol{x}. \tag{7}$$

It is important to acknowledge that matching a finite number of moments does not necessarily lead to the true density ratio, even in asymptotic scenarios. Kernel mean matching (Huang et al., 2006; Gretton et al., 2009) offers a solution by efficiently aligning all moments within the Gaussian Reproducing Kernel Hilbert Space (RKHS) $\mathfrak{H}$:

$$\min_{\hat{r} \in \mathfrak{H}} \left\| \int K(\boldsymbol{x}, \cdot)\hat{r}(\boldsymbol{x})p_{tr}(\boldsymbol{x})d\boldsymbol{x} - \int K(\boldsymbol{x}, \cdot)p_{tr}(\boldsymbol{x})d\boldsymbol{x} \right\|_{\mathfrak{H}}^2. \tag{8}$$

Kullback–Leibler Importance Estimation Procedure (KLIEP) (Nguyen et al., 2010; Sugiyama et al., 2007b) minimizes KL-divergence from $p_{te}(\boldsymbol{x})$ to $\hat{p}_{te}(\boldsymbol{x}) = \hat{r}(\boldsymbol{x})p_{tr}(\boldsymbol{x})$ as

$$\min_{\hat{r}} D_{KL}[p_{te}(\boldsymbol{x}) \| \hat{p}_{te}] = \min_{\hat{r}} \int p_{te}(\boldsymbol{x}) \frac{p_{te}(\boldsymbol{x})}{\hat{r}(\boldsymbol{x})p_{tr}(\boldsymbol{x})} d\boldsymbol{x}.$$

Least-Squares Importance Fitting (LSIF) (Kanamori et al., 2009) minimizes squared loss:

$$\min_{\hat{r}} \int \left(\hat{r}(\boldsymbol{x}) - r(\boldsymbol{x})\right)^2 p_{tr}(\boldsymbol{x})d\boldsymbol{x}.$$

The estimated density ratio obtained in these manners plays a central role in the importance of weighting-based methods. When the disparity between the two distributions is substantial, the accuracy of density ratio estimation significantly deteriorates. Rhodes et al. (2020) propose Telescoping Density Ratio Estimation (TRE) following a divide-and-conquer framework. TRE initially generates a chain of intermediate datasets between two distributions, gradually transporting samples from the source distribution $p_0$ to the target distribution $q = p_m$:

$$\frac{p_0(\boldsymbol{x})}{p_m(\boldsymbol{x})} = \frac{p_0(\boldsymbol{x})p_1(\boldsymbol{x})}{p_1(\boldsymbol{x})p_2(\boldsymbol{x})} \cdots \frac{p_{m-2}(\boldsymbol{x})p_{m-1}(\boldsymbol{x})}{p_{m-1}(\boldsymbol{x})p_m(\boldsymbol{x})}. \tag{9}$$

Experimental results have demonstrated the effectiveness of density ratio estimation along this chain. Moreover, Yin (2023) revealed several statistical properties of TRE. Featurized Density Ratio Estimation (FDRE) (Choi et al., 2021) considers the use of invertible generative models to map two distributions to a common feature space for density ratio estimation. In this feature space, the two distributions become closer to each other in a way that allows for easy calculation of the density ratio. Furthermore, because the generative model used for mapping is invertible, it guarantees that the density ratio calculated in this common space is identical in the input space as well. Trimmed Density Ratio Estimation (TDRE) (Liu et al., 2017) is a robust method that automatically detects outliers, which can cause density ratio estimation values to become large, and mitigate their influence. Kumagai et al. (2021) utilize a meta-learning framework for relative density ratio estimation.

As we will see later, much of the discussion about importance weighting in machine learning is strongly related to density ratios. Therefore, one can say that density ratio estimation is a foundamental technique that supports importance weighting.

## 3 Distribution Shift Adaptation

Many supervised learning algorithms assume that training and test data are generated from the identical distribution, as $p(\boldsymbol{x}, y) = p_{tr}(\boldsymbol{x}, y) = p_{te}(\boldsymbol{x}, y)$. However, this assumption is often violated. This problem setting is called the dataset shift or distribution shift. In this section, we introduce how the importance weighting methods are utilized to tackle the distribution shift assumption.

### 3.1 Covariate Shift

The covariate shift assumption was reported by Shimodaira (2000) and is formulated as follows.

**Definition 3** (Covariate Shift (Shimodaira, 2000))**.** When distributions $p_{tr}(\boldsymbol{x}, y)$ and $p_{te}(\boldsymbol{x}, y)$ satisfy the following condition, it is said that the covariate shift assumption holds:

$$p_{tr}(\boldsymbol{x}) \neq p_{te}(\boldsymbol{x}),$$
$$p_{tr}(y|\boldsymbol{x}) = p_{te}(y|\boldsymbol{x}) = p(y|\boldsymbol{x}).$$

Under this assumption, we can see that the unbiasedness of estimator obtained by Empirical Risk Minimization is not satisfied. This issue is known to be solvable by utilizing the density ratio between the training distribution and the testing distribution, as follows:

$$
\begin{aligned}
\mathbb{E}_{p_{tr}(\boldsymbol{x}, y)} \left[ \frac{p_{te}(\boldsymbol{x})}{p_{tr}(\boldsymbol{x})} \ell(h(\boldsymbol{x}), y) \right] &= \int_{\mathcal{X} \times \mathcal{Y}} \frac{p_{te}(\boldsymbol{x})}{p_{tr}(\boldsymbol{x})} \ell(h(\boldsymbol{x}), y) \cdot p_{tr}(\boldsymbol{x}, y) d\boldsymbol{x} dy \\
&= \int_{\mathcal{X} \times \mathcal{Y}} \frac{p_{te}(\boldsymbol{x})}{p_{tr}(\boldsymbol{x})} \ell(h(\boldsymbol{x}), y) \cdot p_{tr}(\boldsymbol{x}) p_{tr}(y|\boldsymbol{x}) d\boldsymbol{x} dy \\
&= \int_{\mathcal{X} \times \mathcal{Y}} p_{te}(\boldsymbol{x}) \ell(h(\boldsymbol{x}), y) \cdot p_{tr}(y|\boldsymbol{x}) d\boldsymbol{x} dy \\
&= \int_{\mathcal{X} \times \mathcal{Y}} \ell(h(\boldsymbol{x}), y) \cdot p_{te}(\boldsymbol{x}, y) d\boldsymbol{x} dy = \mathbb{E}_{p_{te}(\boldsymbol{x}, y)} \left[ \ell(h(\boldsymbol{x}), y) \right]. \quad (10)
\end{aligned}
$$

Minimization of the weighted loss in Eq. 10 is called the Importance Weighted Empirical Risk Minimization (IWERM) (Shimodaira, 2000). IWERM is theoretically valid, but it has been reported to be unstable in numerical experiments. To stabilize IWERM, Adaptive Importance Weighted Empirical Risk Minimization (AIWERM) (Shimodaira, 2000), which replaces importance weighting with $\left( \frac{p_{te}(\boldsymbol{x})}{p_{tr}(\boldsymbol{x})} \right)^{\lambda}$, $\lambda \in [0, 1]$, has been proposed, and its effectiveness has been demonstrated. Relative Importance Weighted Empirical Risk Minimization (RIWRM) (Yamada et al., 2013) is another alternative to IWERM, achieving stability by using $\left( \frac{p_{te}(\boldsymbol{x})}{(1-\lambda)p_{tr}(\boldsymbol{x}) + \lambda p_{te}(\boldsymbol{x})} \right)$ as importance weights, where $\lambda \in [0, 1]$. A comparison of IWERM, AIWERM and RIWERM is shown in Table 1. Moreover, it is demonstrated that IWERM, AIWERM and RIWERM are generalized in terms of information geometry (Kimura & Hino, 2022).

Table 1: Comparison of IWERM, AIWERM, and RIWERM. IWERM is statistically unbiased and behaves well when the sample size is large enough, while it is numerically unstable when the sample size is small. On the other hand, AIWERM and RIWERM gain robustness by allowing for bias in estimators. It is also known that RIWERM is computationally more stable because IWERM and AIWERM include the density ratio in the calculation of the weight function, while RIWERM replaces the denominator with a weighted average of the two distributions.

| Algorithm | Weighting function $w(\boldsymbol{x})$ | Parameters | Pros (+) & Cons (−) |
|---|---|---|---|
| IWERM (Shimodaira, 2000) | $p_{te}(\boldsymbol{x})/p_{tr}(\boldsymbol{x})$ | - | + Unbiased estimators 
 + No tuning parameters 
 − Numerically unstable |
| AIWERM (Shimodaira, 2000) | $\{p_{te}(\boldsymbol{x})/p_{tr}(\boldsymbol{x})\}^{\lambda}$ | $\lambda \in [0,1]$ | + Robust estimators 
 − Biased estimators |
| RIWERM (Yamada et al., 2013) | $p_{te}(\boldsymbol{x})/\{(1-\lambda)p_{tr}(\boldsymbol{x}) + \lambda p_{te}(\boldsymbol{x})\}$ | $\lambda \in [0,1]$ | − Tuning parameter $\lambda$ |

Importance weighting is also known to be effective in model selection. Cross-validation is an essential and practical strategy for model selection but is subject to bias under covariate shift. We define the error

$$\mathcal{R}^{(n)} := \mathbb{E}_{S_n,(\boldsymbol{x},\boldsymbol{y})}\left[\ell(h_{S_n}(\boldsymbol{x}),y)\right], \tag{11}$$

of a hypothesis $h_{S_n}$ trained by a sample $S_n$ of size $n$. The Leave-One-Out Cross Validation (LOOCV) estimate $\hat{\mathcal{R}}^{(n)}_{\text{LOOCV}}$ of the error $\mathcal{R}^{(n)}$ is

$$\hat{\mathcal{R}}^{(n)}_{\text{LOOCV}} := \frac{1}{n}\sum_{j=1}^{n}\ell(h_{S_n^j}(\boldsymbol{x}_j),y_j), \tag{12}$$

where $S_n^j = \{(\boldsymbol{x}_i,y_i)\}_{i\neq j}^n$. It is known that, if $p_{tr} = p_{te}$, LOOCV gives an almost unbiased estimate of the risk (Luntz, 1969; Smola & Schölkopf, 1998) as

$$\mathbb{E}_{S_n}\left[\hat{\mathcal{R}}^{(n)}_{\text{LOOCV}}\right] = \mathcal{R}^{(n-1)} \approx \mathcal{R}^{(n)}. \tag{13}$$

However, this property is no longer true under covariate shift. Importance Weighted Cross Validation (IWCV) (Sugiyama et al., 2007a), a variant of the cross-validation that applies importance weighting, overcomes this problem. The Leave-One-Out Importance-Weighted Cross Validation (LOOIWCV) is defined as

$$\hat{\mathcal{R}}^{(n)}_{\text{LOOCV}} := \frac{1}{n}\sum_{j=1}^{n}\frac{p_{te}(\boldsymbol{x}_j)}{p_{tr}(\boldsymbol{x}_j)}\ell(h_{S_n^j}(\boldsymbol{x}_j),y_j). \tag{14}$$

We can see that the validation error in the cross-validation procedure is weighted according to the importance of input vectors. For any $j \in \{1,\ldots,n\}$, we have

$$\mathbb{E}_{S_n}\left[\frac{p_{te}(\boldsymbol{x}_j)}{p_{tr}(\boldsymbol{x}_j)}\ell(h_{S_n^j}(\boldsymbol{x}_j),y_j)\right] = \mathbb{E}_{S_n^j,\boldsymbol{x}_j,y_j}\left[\frac{p_{te}(\boldsymbol{x}_j)}{p_{tr}(\boldsymbol{x}_j)}\ell(h_{S_n^j}(\boldsymbol{x}_j),y_j)\right]$$

$$= \mathbb{E}_{S_n^j,y_j}\left[\int_{\mathcal{X}}\frac{p_{te}(\boldsymbol{x}_j)}{p_{tr}(\boldsymbol{x}_j)}\ell(h_{S_n^j}(\boldsymbol{x}_j),y_j)p_{tr}(\boldsymbol{x}_j)d\boldsymbol{x}_j\right]$$

$$= \mathbb{E}_{S_n^j,y_j}\left[\int_{\mathcal{X}}p_{te}(\boldsymbol{x}_j)\ell(h_{S_n^j}(\boldsymbol{x}_j),y_j)d\boldsymbol{x}_j\right]$$

$$= \mathbb{E}_{S_n^j,y}\left[\int_{\mathcal{X}}p_{te}(\boldsymbol{x})\ell(h_{S_n^j}(\boldsymbol{x}),y)d\boldsymbol{x}\right] = \mathbb{E}_{S_n^j,\boldsymbol{x},y}\left[\ell(h_{S_n^j}(\boldsymbol{x}),y)\right] = \mathcal{R}^{(n-1)} \approx \mathcal{R}^{(n)},$$

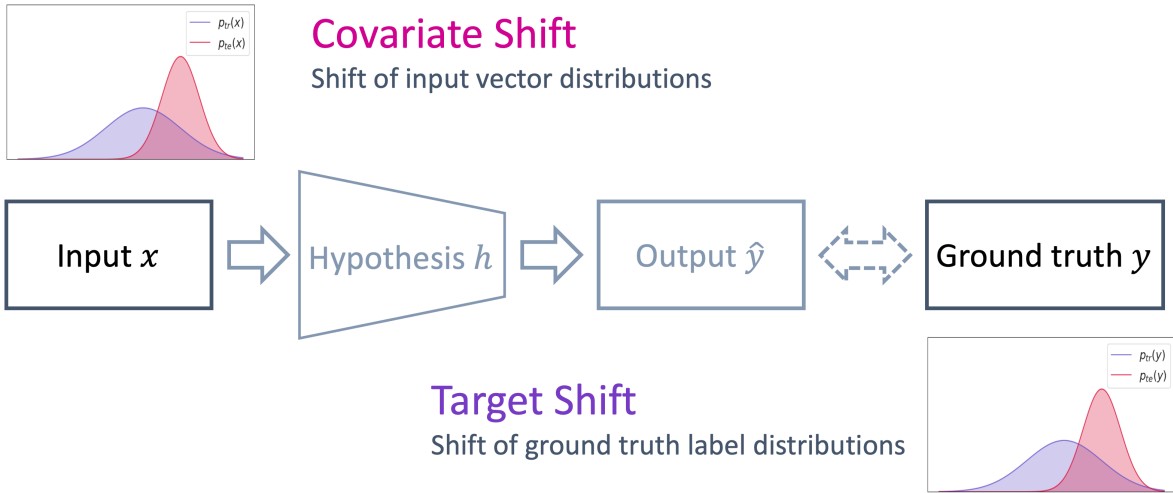

Figure 2: Illustration of two distribution shifts. Covariate shift assumes a shift in the distribution of the input vector $\boldsymbol{x}$ and target shift assumes a shift in the distribution of the output $y$.

and

$$
\begin{aligned}
\mathbb{E}_{S_n}\left[\hat{\mathcal{R}}_{\text{LOOIWCV}}^{(n-1)}\right] &= \mathbb{E}_{S_n}\left[\frac{1}{n}\sum_{j=1}^{n}\frac{p_{te}(\boldsymbol{x}_j)}{p_{tr}(\boldsymbol{x}_j)}\ell(h_{S_n^j}(\boldsymbol{x}_j), y_j)\right] \\
&= \frac{1}{n}\sum_{j=1}^{n}\mathbb{E}_{S_n}\left[\frac{p_{te}(\boldsymbol{x}_j)}{p_{tr}(\boldsymbol{x}_j)}\ell(h_{S_n^j}(\boldsymbol{x}_j), y_j)\right] = \frac{1}{n}\sum_{j=1}^{n}\mathcal{R}^{(n-1)} = \mathcal{R}^{(n-1)} \approx \mathcal{R}^{(n)}.
\end{aligned}
$$

Subsequently, we observe that LOOIWCV provides an almost unbiased estimate of the error, even in the presence of covariate shift. This proof can be readily extended to include an importance-weighted version of k-fold cross validation.

However, Kouw & Loog (2016) reported their experimental findings, indicating various estimates of IWERM consistently underestimated the expected error. Moreover, classical results have highlighted the benefits of importance weighting specifically when the model is parametric. Gogolashvili et al. (2023) demonstrated the necessity of importance weighting, even in scenarios where the model is non-parametric and misspecified.

### 3.2 Target Shift

In the context of target shift (Zhang et al., 2013), it is assumed that the target marginal distribution undergoes changes between the training and testing phases, while the distribution of covariates conditioned on the target remains unchanged.

**Definition 4** (Target Shift (Zhang et al., 2013))**.** When distributions $p_{tr}(\boldsymbol{x}, y)$ and $p_{te}(\boldsymbol{x}, y)$ satisfy the following condition, it is said that the target shift assumption holds:

$$
\begin{aligned}
p_{tr}(y) &\neq p_{te}(y), \\
p_{tr}(\boldsymbol{x}|y) &= p_{te}(\boldsymbol{x}|y) = p(\boldsymbol{x}|y).
\end{aligned}
$$

Figure 2 shows a conceptual illustration of covariate shift and target shift. In cases where the target variable is continuous, additional complexities arise, leading to new research opportunities. Chan & Ng (2005) employ the Expectation-Maximization (EM) method to infer the distribution $p(y)$, which also entails estimating $p(\boldsymbol{x}|y)$. Nguyen et al. (2016) introduce a technique for calculating importance weights $p_{te}(y)/p_{tr}(y)$ to tackle the shift in continuous target variables within a semi-supervised framework. Black Box Shift Estimation (BBSE) (Lipton et al., 2018) estimates importance weights by solving the equation $\boldsymbol{C}\boldsymbol{w} = b$ through the

confusion matrix $C$ of a black box predictor $f$ (where $b$ denotes the predictor's average output). Lipton et al. (2018) show that the error bound for the weighting $\hat{w}$ estimated by BBSE under appropriate assumptions can be derived as

$$\|\hat{w} - w\|_2^2 \leq \frac{C}{\lambda_{\min}^2} \left( \frac{\|w\|^2 \log n}{n} + \frac{k \log m}{m} \right), \tag{15}$$

where $\lambda_{\min}$ is the smallest eigenvalue of $C$, $n$ is the sample size from the training distribution, $m$ is the sample size from the test distribution and $k$ is the number of classes.

### 3.3 Sample Selection Bias

Although it bears a close resemblance to the covariate shift assumption, many studies also address similar problem settings under the umbrella of sample selection bias (Tran, 2017; Winship & Mare, 1992; Vella, 1998; Berk, 1983).

Numerous studies (Zadrozny, 2004; Tran & Aussem, 2015) introduced a random variable $s$ to represent sample selection bias, formulating the test distribution as:

$$p_{te}(x, y) = \sum_s p(x, y, s), \tag{16}$$

where $s = 1$ indicates selection of the example, and $s = 0$ its non-selection. Here, the assumption of the sample selection bias is defined as follows.

**Definition 5** (Sample Selection Bias). Let $P_{tr}(s \mid \boldsymbol{x})$ and $P_{te}(s \mid \boldsymbol{x})$ be the probability of sample selection given $\boldsymbol{x}$ at training and testing phase, respectively. Then, when they satisfy the following condition, it is said that the sample selection bias assumption holds:

$$P_{tr}(s \mid \boldsymbol{x}) \neq P_{te}(s \mid \boldsymbol{x}). \tag{17}$$

For example, in Bayesian classifiers, the posterior probability $P(y \mid \boldsymbol{x})$ for discrete $y$ is written as

$$P(y \mid \boldsymbol{x}) = P(y \mid \boldsymbol{x}, s = 1) = \frac{P(\boldsymbol{x} \mid y, s = 1) P(y \mid s = 1)}{P(\boldsymbol{x} \mid s = 1)}, \tag{18}$$

where $P$ represents the probability of discrete random variables. Based on these assumptions, they illustrate the efficacy of importance weighting, where

$$w(\boldsymbol{x}) = \frac{P(s = 1)}{P(s = 1 \mid \boldsymbol{x})}. \tag{19}$$

The following theorem demonstrates the usefulness of importance weighting in machine learning under sample selection bias.

**Theorem 1** (Zadrozny (2004)). *If we assume that $P(s = 1 \mid x, y) = P(s = 1 \mid x)$, then we have*

$$\mathbb{E}_{tr}\left[w(\boldsymbol{x})\ell(h(\boldsymbol{x}), y) \mid s = 1\right] = \mathbb{E}_{te}\left[\ell(h(\boldsymbol{x}), y)\right], \tag{20}$$

*under the sample selection bias.*

The key calculation is a rewriting of the following conditional expectation with importance weighting.

$$\begin{aligned}
\mathbb{E}_{tr}\left[w(\boldsymbol{x})\ell(h(\boldsymbol{x}), y) \mid s = 1\right] &= \sum_{\boldsymbol{x}, y} w(\boldsymbol{x})\ell(h(\boldsymbol{x}), y) P_{tr}(\boldsymbol{x}, y \mid s = 1) \\
&= \sum_{\boldsymbol{x}, y} \frac{P_{te}(s = 1)}{P_{te}(s = 1 \mid \boldsymbol{x})} \ell(h(\boldsymbol{x}), y) P_{te}(\boldsymbol{x}, y \mid s = 1) \\
&= \sum_{\boldsymbol{x}, y} \ell(h(\boldsymbol{x}), y) P_{te}(\boldsymbol{x}, y, s) \\
&= \mathbb{E}_{te}\left[\ell(h(\boldsymbol{x}), y)\right].
\end{aligned}$$

This means that importance weighting can achieve a correction for sample selection bias.

Nevertheless, in cases with pronounced sample selection bias, Liu & Ziebart (2014) highlight that the errors resulting from importance weighting can become excessively large.

### 3.4 Subpopulation Shift

Subpopulation shift (Santurkar et al., 2020; Yang et al., 2023) in machine learning literature describes a scenario where the statistical characteristics of a specific subset or subpopulation of data undergo changes between the training and testing phases. This implies that the data distribution within a particular subgroup, such as a certain demographic, geographical location, or any other distinct subset, varies between the training and testing datasets. The formal definition of the subpopulation shift is as follows.

**Definition 6** (Subpopulation Shift)**.** Assume that the train and test distributions have the following structure:

$$\mathrm{supp}(p_{tr}(\boldsymbol{x})) = \bigcup_{i=1}^{m} S_i,$$
$$\mathrm{supp}(p_{te}(\boldsymbol{x})) = \bigcup_{i=1}^{m} T_i, \tag{21}$$

where

$$\forall i \neq j, \quad S_i \cap S_j = T_i \cap T_j = S_i \cap T_j = \emptyset. \tag{22}$$

Then, when they satisfy the following condition, it is said that the subpopulation shift assumption holds:

$$\exists i \in [1, m], \quad p_{tr}(S_i) \neq p_{te}(T_i). \tag{23}$$

Numerous studies suggest reweighting sample inversely proportional to the subpopulation frequencies (Cao et al., 2019b; Cui et al., 2019; Liu & Chawla, 2011). As an alternative, UMIX (Han et al., 2022) is a variant of mixup data augmentation technique, designed to calculate importance weights and enhance model performance in the presence of subpopulation shift. The original mixup training (Zhang et al., 2018a; Kimura, 2021; Zhang & Deng, 2021) is a data augmentation in which the weighted average of the two input instances is taken as the new instance:

$$\tilde{\boldsymbol{x}}_{i,j} = \lambda \boldsymbol{x}_i + (1 - \lambda)\boldsymbol{x}_j,$$

and its objective function to be minimized is given as

$$\mathbb{E}_{\{(\boldsymbol{x}_i, y_i), (\boldsymbol{x}_j, y_j)\}} \left[ \lambda \ell(h(\tilde{\boldsymbol{x}}_{i,j}), y_i) + (1 - \lambda)\ell(h(\tilde{\boldsymbol{x}}_{i,j}), y_j) \right],$$

where $\boldsymbol{x}_i, \boldsymbol{x}_j, y_i, y_j$ are the $i$-th and $j$-th input instances and corresponding labels in the training dataset, respectively. UMIX modifies this procedure by using the importance weighting

$$\mathbb{E}_{\{(\boldsymbol{x}_i, y_i), (\boldsymbol{x}_j, y_j)\}} \left[ w(\boldsymbol{x}_i)\lambda \ell(h(\tilde{\boldsymbol{x}}_{i,j}), y_i) + w(\boldsymbol{x}_j)(1 - \lambda)\ell(h(\tilde{\boldsymbol{x}}_{i,j}), y_j) \right],$$

where $w(\boldsymbol{x}_i)$ is obtained by the uncertainty of a classifier $h$.

### 3.5 Feedback Shift

In the advertising industry, accurately predicting the number of clicks leading to purchases is vital. However, it is widely recognized that a time lag exists between clicks and actual purchases in practical scenarios. This issue is commonly referred to as feedback shift or delayed feedback. Delayed feedback was initially researched by Chapelle (2014) and has since led to the proposal of numerous techniques to address this problem (Yoshikawa & Imai, 2018; Ktena et al., 2019; Safari et al., 2017; Tallis & Yadav, 2018). Additionally, the concept of delayed feedback has been examined in the context of bandit problems (Joulani et al., 2013; Pike-Burke et al., 2018; Vernade et al., 2017). The formal definition of the feedback shift is as follows.

**Definition 7** (Feedback Shift)**.** Let $s \in \{0, 1\}$ be a random variable indicating whether a sample is correctly labeled if $s = 1$ in the training data, $d \in \mathbb{R}$ be a random variable of the delay which is a gap between a sample acquisition and its feedback, and $e \in \mathbb{R}$ be a random variable of the elapsed time between a sample acquisition and its labeling. If some positive samples are mislabeled ($s = 0$) when their elapsed time $e$ is shorter than their delay $d$ as

$$e < d \Rightarrow s = 0,$$

then it is said that the feedback shift assumption holds.

Owing to feedback shift, there are instances where data that should be labeled positively is incorrectly labeled negatively. For instance, due to the delay in feedback, purchases might not have occurred at the time of collecting training instances. To address this issue, Yasui et al. (2020) utilize importance weighting, termed feedback shift importance weighting (FSIW). Let $C$ be a $\{0, 1\}$-valued random variable indicating whether a conversion occurs if $C = 1$. In addition, let $Y$ be a $\{0, 1\}$-valued random variable indicating whether a conversion occurs during the training term if $Y = 1$. Then, applying importance weighting $w(\boldsymbol{x}_i) = \frac{P(C=y_i|X=\boldsymbol{x}_i)}{P(Y=y_i|X=\boldsymbol{x}_i)}$ yields

$$
\begin{aligned}
\mathbb{E}_{(\boldsymbol{x},y)} \left[ \frac{P(C=y \mid X=\boldsymbol{x})}{P(Y=y \mid X=\boldsymbol{x})} \ell(h(\boldsymbol{x}), y) \right] &= \sum_{\boldsymbol{x} \in \mathcal{X}} \sum_{y \in \{0,1\}} \frac{P(C=y \mid X=\boldsymbol{x})}{P(Y=y \mid X=\boldsymbol{x})} \ell(h(\boldsymbol{x}), y) P(X=\boldsymbol{x}, Y=y) \\
&= \sum_{\boldsymbol{x} \in \mathcal{X}} \sum_{y \in \{0,1\}} \frac{P(C=y \mid X=\boldsymbol{x})}{P(Y=y \mid X=\boldsymbol{x})} \ell(h(\boldsymbol{x}), y) P(X=\boldsymbol{x}) P(Y=y \mid X=\boldsymbol{x}) \\
&= \sum_{\boldsymbol{x} \in \mathcal{X}} \sum_{y \in \{0,1\}} P(C=y \mid X=\boldsymbol{x}) \ell(h(\boldsymbol{x}), y) P(X=\boldsymbol{x}) \\
&= \sum_{\boldsymbol{x} \in \mathcal{X}} \sum_{c \in \{0,1\}} \ell(h(\boldsymbol{x}), c) P(C=c \mid X=\boldsymbol{x}) P(X=\boldsymbol{x}) = \mathbb{E}_{(\boldsymbol{x},c)} \left[ \ell(h(\boldsymbol{x}), c) \right].
\end{aligned}
$$

This suggests that FSIW leads the consistent estimator under the feedback shift.

### 3.6 Domain Adaptation

Domain adaptation (Patel et al., 2015; Ben-David et al., 2006; Wang & Deng, 2018; Farahani et al., 2021; Csurka, 2017; Wilson & Cook, 2020) in machine learning encompasses the adaptation of a model trained on one domain (the source domain) to accurately predict in a different, yet often related, domain (the target domain). Domains may exhibit variations in aspects such as data distribution, feature representations, or underlying data generation processes.

We borrow the definition of domain adaptation from the book by Redko et al. (2019).

**Definition 8** (Domain Adaptation)**.** Let $\mathcal{X}_S \times \mathcal{Y}_S$ be the source input and output spaces associated with a source distribution $p_S$, and $\mathcal{X}_T \times \mathcal{Y}_T$ be the target input and output spaces associated with a target distribution $T$. We denote $p_S(x)$ and $p_T(x)$ the marginal distributions of $\mathcal{X}_S$ and $\mathcal{X}_T$. Then, domain adaptation aims to improve the learning of the target predictive function $h_T \colon \mathcal{X}_T \to \mathcal{Y}_T$ using knowledge gained from $p_S$ where $p_S(x, y) \neq p_T(x, y)$.

The difference between the above definition and covariate shift or target shift is whether or not the conditional identity of the source and target distributions is assumed. In fact, while Definitions 3 and 4 assume that $p_{tr}(\boldsymbol{x} \mid y) = p_{te}(\boldsymbol{x} \mid y)$ or $p_{tr}(y \mid \boldsymbol{x}) = p_{te}(y \mid \boldsymbol{x})$, respectively, the domain adaptation problem setting makes no such assumptions about the conditional distribution.

Significant research focuses on the importance of weighting for domain adaptation. A key concept among these studies is the allocation of higher weights to source domain data more similar to the target domain. Lee et al. (2018) suggested a technique that approximates importance weighting using a distance kernel for domain adaptation. Mashayekhi (2022) developed a method to estimate importance weighting through adversarial learning, in which a generator is trained to assign weights to training instances to mislead a

discriminator. Xiao & Zhang (2021) also introduced importance weighting that considers the difference in sample sizes between the source and the target domains. Azizzadenesheli et al. (2019) established a generalization error bound for domain adaptation via importance weighting.

Jiang & Zhai (2007) employed importance weighting for domain adaptation in NLP tasks. Nevertheless, some studies in NLP have reported unsatisfactory results with domain adaptation based on importance weighting (Plank et al., 2014). This might be attributed to the specific characteristics of NLP data, such as word occurrence rates. Acknowledging these limitations, Xia et al. (2018) observed that existing strategies primarily address sample selection bias but fall short in handling sample selection variance. To address this issue, they proposed several weighting methods that take into account the trade-off between bias and variance.

Furthermore, domain adaptation encompasses several subfields, such as:

- Multi-source domain adaptation: involving the use of multiple source domains.

- Partial domain adaptation: where the target domain features fewer classes than the source domain.

- Open-set domain adaptation: characterized by unknown classes in both domains.

- Universal domain adaptation: which operates without prior knowledge of the label sets.

### 3.6.1 Multi-Source Domain Adaptation

The objective of multi-source domain adaptation is to utilize the information from the multiple source domains to adapt the model to perform well on the target domain, despite potential differences in data distribution. Sun et al. (2011) introduced a two-stage multi-source domain adaptation framework designed to calculate importance weights for instances from multiple sources, aiming to minimize both marginal and conditional probability discrepancies between the source and target domains. Wen et al. (2020) formulated an algorithm capable of autonomously determining the importance weights of data from various source domains. Shui et al. (2021) also propose to assign different importance weights to each source distribution depending on the similarity between them and the sample size obtained from them.

### 3.6.2 Partial Domain Adaptation

Traditional domain adaptation methods mitigate domain discrepancies in three ways: by developing feature representations invariant across domains, by generating features or samples tailored for target domains, or by transforming samples between domains using generative models. These approaches assume identical label sets across domains. Partial domain adaptation challenges this by assuming that the target domain *comprises fewer classes than* the source domain (Cao et al., 2019b; Liang et al., 2020; Cao et al., 2018b). In partial domain adaptation, the class space of the target domain is a subset of the source domain class space. Consequently, aligning the entire source domain distribution $p_{tr}$ with the target domain distribution $p_{te}$ is problematic. In addition, based on the discussion by Shui et al. (2021), partial domain adaptation can be regarded as a special case of multi-source domain adaptation. In fact, in multi-source domain adaptation, we can recover the problem setting of partial domain adaptation by considering the case where the weight coefficients of some source distributions are zero.

Zhang et al. (2018b) suggested that incorporating importance weighting into adversarial nets-based domain adaptation (IWAN) is effective for the partial domain adaptation problem. Domain adaptation methods based on adversarial neural networks (Creswell et al., 2018; Goodfellow et al., 2020) utilize a discriminator to classify whether input examples originate from real or generative distributions (Long et al., 2018; Tzeng et al., 2017). The weighting function of IWAN is formulated as:

$$w(\boldsymbol{x}) = 1 - D^*(\boldsymbol{x}) = \frac{1}{p_{tr}(\boldsymbol{x})/p_{te}(\boldsymbol{x}) + 1}, \tag{24}$$

where $D^*(\boldsymbol{x})$ denotes the optimal discriminator, $p_{tr}(\boldsymbol{x})$ the source distribution, and $p_{te}(\boldsymbol{x})$ the target distribution. This weighting function resembles the unnormalized version of relative importance weighting (Yamada

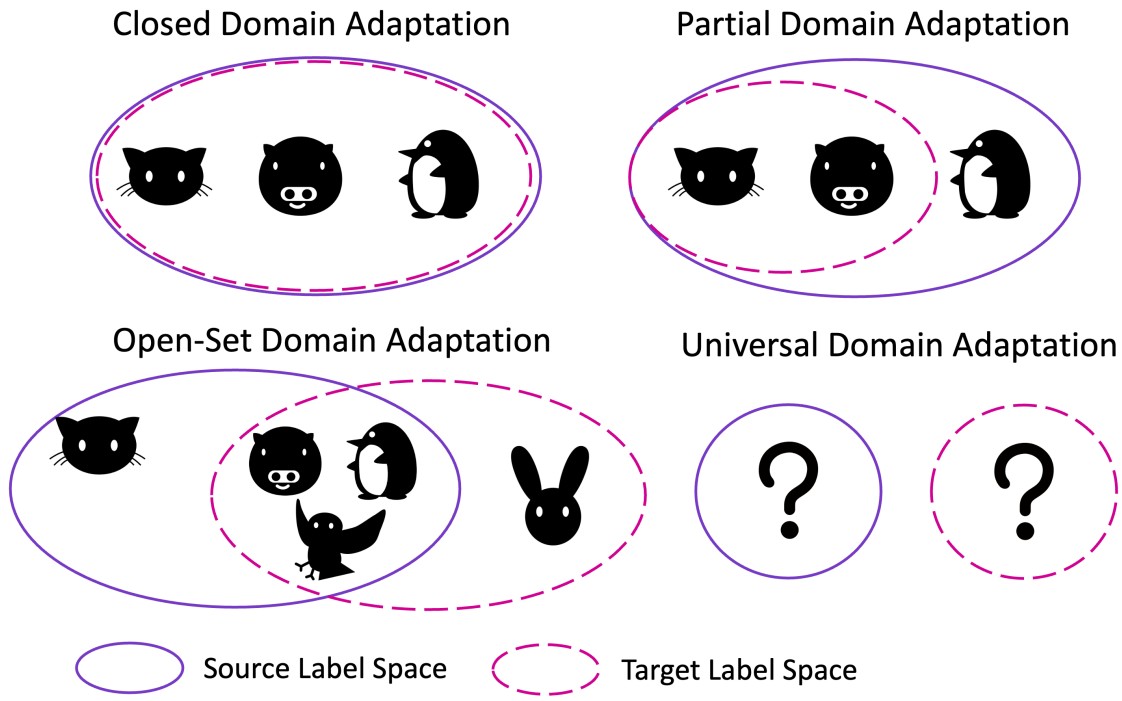

Figure 3: Illustration of different domain adaptation settings.

et al., 2013):

$$1 - D^*(\boldsymbol{x}) = \frac{1}{p_{tr}(\boldsymbol{x})/p_{te}(\boldsymbol{x}) + 1} = \frac{p_{te}(\boldsymbol{x})}{p_{te}(\boldsymbol{x})} \cdot \frac{1}{p_{tr}(\boldsymbol{x})/p_{te}(\boldsymbol{x}) + 1} = \frac{p_{te}(\boldsymbol{x})}{p_{tr}(\boldsymbol{x}) + p_{te}(\boldsymbol{x})}.$$

Example Transfer Network (ETN) (Cao et al., 2019a) extends IWAN by jointly optimizing domain-invariant features and importance weights for relevant source instances. Selective Adversarial Network (SAN) (Cao et al., 2018a) implements multiple adversarial networks with an importance weighting mechanism to filter outsource examples in outlier classes. Furthermore, an extension of IWAN for regression and general domain adaptation is proposed (de Mathelin et al., 2021).

### 3.6.3   Open-Set Domain Adaptation

Recent studies aim to relax the constraint of having identical label sets across domains by introducing open-set domain adaptation. Panareda Busto & Gall (2017) incorporated unknown classes in both domains, operating under the assumption that shared classes between the two domains are identified during training. Formally, the open-set domain adaptation problem is defined as follows.

**Definition 9** (Open-Set Domain Adaptation). Let $\mathcal{Y}_S$ and $\mathcal{Y}_T$ be the label spaces of the source and target distributions, respectively. The open-set domain adaptation is a problem that aims to obtain a good predictor under the following assumptions.

$$\mathcal{Y}_S \neq \mathcal{Y}_T,$$
$$\mathcal{Y}_S \cap \mathcal{Y}_T \neq \emptyset. \tag{25}$$

Liu et al. (2019) emphasized the significance of considering the openness of the target domain, defined as the proportion of unknown classes relative to all target classes. Their methodology employs a coarse-to-fine weighting mechanism to progressively differentiate between samples of unknown and known classes, while emphasizing their contribution to aligning feature distributions. Kundu et al. (2020) introduced a metric termed *model inheritability* to evaluate the effectiveness of a model trained on the source domain when

applied to the target domain. They advocate employing this metric in importance weighting for open-set domain adaptation. Garg et al. (2022) further developed a method for open-set domain adaptation using the concept of Positive-Unlabeled (PU) learning (Bekker & Davis, 2020; Kiryo et al., 2017).

### 3.6.4 Universal Domain Adaptation

Universal domain adaptation is the most challenging problem setting in domain adaptation as it requires no prior knowledge about the label sets in the target domain. Figure 3 shows the illustration of different domain adaptation settings. You et al. (2019) have proposed the Universal Adaptation Network (UAN), which performs instance weighting based on domain similarity and prediction uncertainty. Fu et al. (2020) leverage model calibration techniques to enhance the effectiveness of importance weighting on source domain data based on the uncertainty in universal domain adaptation. They utilize multiple indicators of uncertainty (entropy, confidence and consistency) to define the weighting function.

### 3.7 Distributionally Robust Optimization

Distributionally Robust Optimization (DRO) (Rahimian & Mehrotra, 2019; Goh & Sim, 2010; Duchi & Namkoong, 2018; Bertsimas et al., 2019; Levy et al., 2020; Delage & Ye, 2010) focuses on minimizing the expected loss under the most challenging conditions, considering a probabilistic uncertainty set $\mathcal{U}(p_0)$ inferred from observed samples and characterized by certain attributes of the actual data generation distribution. Specifically, the objective of the DRO problem is

$$\text{minimize}_{h \in \mathcal{H}} \ \mathcal{R}(h; p_0) := \sup_{q \in \mathcal{U}(p_0)} \mathbb{E}_{(\boldsymbol{x}, y) \sim q(\boldsymbol{x}, y)} \Big[ \ell(h(\boldsymbol{x}), y) \Big], \tag{26}$$

where $\mathcal{H}$ *is the hypothesis class and* $\mathcal{U}(p_0)$ is the uncertainty set associated with the training distribution $p_0$. The field encompasses various uncertainty sets (Ben-Tal et al., 2013; Bertsimas et al., 2018; Blanchet et al., 2019; Esfahani & Kuhn, 2015), with the goal of developing models that maintain robust performance amidst perturbations in the data-generating distribution. DRO can also be interpreted as accounting for a worst-case scenario in the face of unknown distribution shifts.

One way to interpret the worst-case formulation in DRO is as importance-weighted loss minimization without specifying a target domain. Utilizing a set of weighting functions $W$, the uncertainty set can be constructed as

$$\mathcal{U}_w(p_0) = \{ w(\boldsymbol{x}) \cdot p_0(\boldsymbol{x}); w \in W \}. \tag{27}$$

Duchi & Namkoong (2021) elaborated on the DRO framework by amplifying the tail values of $\ell(h(\boldsymbol{x}), y)$ resulting in a worst-case objective that concentrates on challenging regions within the input domain $\mathcal{X}$. Awad & Atia (2022) devised an uncertainty set centered around the importance-weighted source domain distribution, showing its high likelihood of covering the true target domain distribution. As a practical optimization method for deep neural network models under data imbalance, a variant of momentum SGD called Attentional-Biased Stochastic Gradient Descent (ABSGD) (Qi et al., 2022) has been proposed. AB-SGD implements importance weighting for each sample in a mini-batch, benefiting from the theoretical underpinnings of DRO.

Additionally, Agarwal & Zhang (2022) have introduced Minimax Regret Optimization (MRO) as a variant of the DRO problem formulation. They have proven that, for both DRO and MRO, the upper bound of their error is determined by the variance of the weighting.

Contrasting with IWERM, which applies weights to the training data using the density ratio $p_{te}(\boldsymbol{x})/p_{tr}(\boldsymbol{x})$, robust algorithms–originating from Distributionally Robust Optimization–assign weights to the test data using $p_{tr}(\boldsymbol{x})/p_{te}(\boldsymbol{x})$ (Liu & Ziebart, 2014; 2017; Chen et al., 2016). Martin et al. (2023) introduced Double-Weighting Covariate Shift Adaptation incorporating a balance between these two weighting approaches.

## 4 Active Learning

Active learning (Settles, 2009; Hino, 2020) is founded on the principle that a learner, by selecting specific instances for labeling, can reduce the necessary sample size to achieve adequate performance. Within active learning, methods vary ranging from those based on uncertainty (Lewis & Catlett, 1994; Yang & Loog, 2016; Zhu et al., 2009; Li & Sethi, 2006), diversity (Brinker, 2003; Yang et al., 2015; Agarwal et al., 2020), representativeness (Du et al., 2015; Huang et al., 2010; Kimura et al., 2020), reducing expected error (Mussmann et al., 2022; Jun & Nowak, 2016), to maximizing expected model changes (Cai et al., 2013; Freytag et al., 2014; Käding et al., 2016). A particular focus within these is on density ratios and importance weighting.

Importance Weighted Active Learning (IWAL) (Beygelzimer et al., 2009) is a pioneering approach that incorporates importance weighting into active learning. IWAL determines whether to label unlabeled instances $\boldsymbol{x}_t$ based on their features and the history of labeled data, with a probability $p_t$. The weight of the instance is then regarded as $1/p_t$ during the learning. The probability $p_t$ is calculated using the hypothesis space $\mathcal{H}_t$, which includes the most contrasting functions $f$ and $g$ in terms of training data accuracy at time $t$. Specifically,

$$p_t = \max_{f,g \in \mathcal{H}_{t+1}} \max_y \sigma(\ell(f(\boldsymbol{x}_t), y) - \ell(g(\boldsymbol{x}_t), y)), \tag{28}$$

$$\mathcal{H}_{t+1} = \{h \in \mathcal{H}_t; L_t(h) \leq L_t^* + \Delta_t\}, \tag{29}$$

where $L_t(h)$ and $L_t^*$ represents the empirical loss and the minimum achievable loss at time $t$, respectively, and $\Delta_t$ is a small quantity. IWAL has been validated for its consistency and effectiveness through importance weighting. This framework has been reported to be effective in a wide range of applications, including image recognition (Wu et al., 2011), point cloud segmentation (Wu et al., 2021), sleep monitoring (Hossain et al., 2015), and battery health management (Chen & Li, 2011). Beygelzimer et al. (2011) introduced practical implementations of IWAL by setting a rejection threshold based on a deviation bound, indicating that IWAL can produce reusable samples. Here, sample reusability (Tomanek, 2010; Tomanek & Morik, 2011) examines if a labeled sample from one classifier can train another in an active learning setup. Contrarily, Van Tulder (2012) reported the infeasibility of generating reusable samples in IWAL in their subsequent work.

Regarding model misspecification, similar issues have been explored. Sugiyama (2005) found that active learning methods can endure minor model misspecifications and that importance weighting can address larger discrepancies. Bach (2006) investigated the asymptotic properties of active learning in generalized linear models under model misspecification and proposed efficient instance selection based on importance weighting according to their findings.

Active Learning by Learning (ALBL) (Hsu & Lin, 2015) combines multiple instance selection algorithms in active learning using a multi-armed bandit framework, selecting the most suitable for a given task. ALBL enhances IWAL by introducing Importance Weighted Accuracy as a reward function for the embedded multi-armed bandit algorithm.

One common challenge in active learning is the difficulty of performance evaluation. Active learning algorithms continuously select highly informative instances for labeling, leading to a labeled dataset whose distribution may diverge from that of the original data. This phenomenon exemplifies sampling bias. Kottke et al. (2019) observed that reweighted cross-validation, a natural extension of IWAL, does not fully rectify these distributional disparities. Zhao et al. (2021) demonstrate that combining importance weighting with class-balanced sampling can reduce sample complexity for arbitrary label shifts. Mohamad et al. (2018) suggested a stream-based active learning method using importance weighting under the concept drift assumption. Su et al. (2020) presented a domain adaptation method merging domain adversarial learning and active learning. In this approach, importance weighting is applied by approximating the density ratio using the output of the discriminator in adversarial learning. Additionally, active testing (Kossen et al., 2021; Farquhar et al., 2021) is a known method for model evaluation using loss-proportional sampling.

Furthermore, in active learning, addressing the variability in annotator quality is paramount. In scenarios like crowdsourcing, where annotators may have varying levels of expertise, addressing the issue of annotator quality is indeed crucial. Zhao et al. (2012) have proposed a pool-based active learning approach based on the concept of IWAL, which is robust to annotation noise.

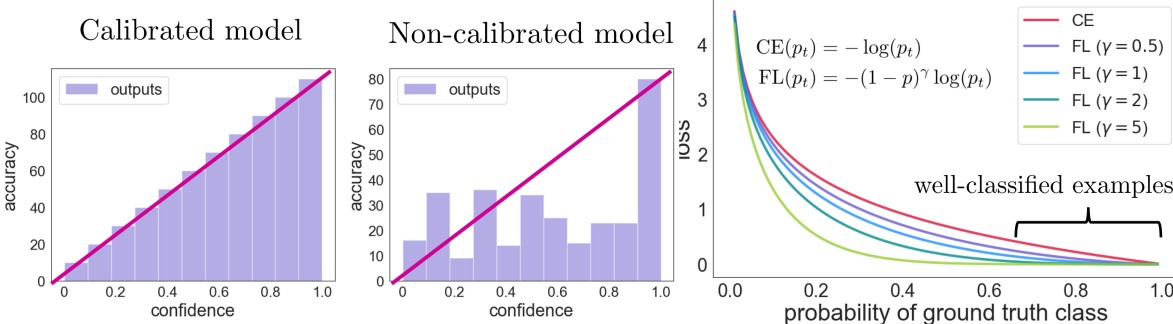

Figure 4: Left panel: the reliability diagrams of two models. Left one is well calibrated than right one. Right panel: illustrative plot of focal loss. By increasing the focal loss parameter $\gamma$, we can see that the well-classified examples are downweighted.

Another challenge in active learning is the issue of computational cost. Agarwal et al. (2013) employed the IWAL framework to parallelize the identification of data needing labeling.

Furthermore, active learning and importance weighting can be related from another perspective. When matching the data distribution estimated on unlabeled (pooled) data with the query distribution in the active learning problem setting such as Shui et al. (2020), based on Rényi-divergence, the expected value weighted by the density ratio appears and this indicates that importance weighting is still important.

## 5 Model Calibration

In many machine learning models, output probabilities are often equated with confidence in their predictions. However, these probabilities may not be well-calibrated, indicating a discrepancy between them and the actual likelihood of predicted outcomes. For example, a model might assign a probability of 0.7 to an event, whereas the real occurrence rate of that event might be approximately 0.9. Model calibration typically involves modifying the output probabilities using various methods, aligning the predicted probabilities with the observed event frequencies in the data (Hill, 2000; Vaicenavicius et al., 2019; Beven & Binley, 1992; Gupta et al., 2006; Eckhardt et al., 2005).

Focal loss is one of the most prominent methods for model calibration. Originally developed for object detection in computer vision (Lin et al., 2017; Li et al., 2020b; Mulyanto et al., 2020; Tran et al., 2019), focal loss has been validated as effective for model calibration in later studies (Mukhoti et al., 2020). Focal loss addresses the calibration problem by down-weighting the easy-to-classify examples (those with high confidence predictions) during training, effectively giving more emphasis to the difficult-to-classify examples. This is achieved with a modulating factor applied to standard cross-entropy loss. This approach can be seen as applying an importance weighting $w(\boldsymbol{x}_i) = (1 - p_i)^\gamma$ that depends on the predictive probabilities $p_i$ of the model for instance $\boldsymbol{x}_i$ and $\gamma \in [0, 1]$. Figure 4 shows a graphical example of the model calibration and a plot of the focal loss behavior.

The simplicity of the idea of focal loss has inspired a plethora of variations. Cyclical focal loss (Smith, 2022) is a variant of the focal loss inspired by curriculum learning (Bengio et al., 2009; Wang et al., 2021; Graves et al., 2017; Hacohen & Weinshall, 2019), which applies weighting to each instance based on its classification difficulty during each epoch. AdaFocal (Ghosh et al., 2022) utilizes the calibration properties of focal loss and dynamically modulates the $\gamma$ parameter for different sample groups based on the model's previous performance and its propensity for overconfidence, as observed in the validation set. Dual focal loss (Tao et al., 2023) refines focal loss by striking a balance between overconfidence and underconfidence. Adversarial Focal loss (AFL) (Liu et al., 2022) employs a distinct adversarial network to generate a difficulty score for each input.

Theoretical insights into focal loss have also emerged. Charoenphakdee et al. (2021) proved that the optimal classifier trained with focal loss can achieve the Bayes-optimal classifier. Another study (Naganuma & Kimura, 2023) showed that focal loss reduces the curvature of the loss surface.

# 6 Positive-Unlabelled (PU) learning

PU learning, an acronym for Positive-Unlabeled learning, refers to a type of machine learning paradigm where the goal is to build a classifier when you have a dataset with positive examples and unlabeled examples (Bekker & Davis, 2020; Kiryo et al., 2017). The principal challenge in PU learning is the absence of labeled negative examples, which are typically integral to training binary classifiers.. In this paradigm, the positive class is well-defined, but the negative class is composed of both true negatives and unlabeled examples. The main objective of PU learning is to develop algorithms that can learn a reliable classifier despite the absence of labeled negatives. This is particularly useful in situations where obtaining a representative sample of negatives is difficult or costly.

Elkan & Noto (2008), the pioneers of PU Learning, based their algorithm on the following lemma:

**Lemma 1** ((Elkan & Noto, 2008)). *Let $\boldsymbol{x}$ be an example and let $y \in \{0, 1\}$ be a binary label. Let $s = 1$ if the example $\boldsymbol{x}$ is labeled, and let $s = 0$ if $\boldsymbol{x}$ is unlabeled. Then, for the completely unlabeled random example $\boldsymbol{x}$, we have $P(y = 1|\boldsymbol{x}) = \frac{P(s=1|\boldsymbol{x})}{P(s=1|y=1)}$.*

We demonstrate the relationship between PU learning and importance weighting as follow. For the unlabeled example,

$$P(y = 1|\boldsymbol{x}, s = 0) = \frac{P(s = 0|\boldsymbol{x}, y = 1)P(y = 1|\boldsymbol{x})}{P(s = 0|\boldsymbol{x})} = \frac{(1 - c)P(s = 1|\boldsymbol{x})/c}{1 - P(s = 1|\boldsymbol{x})} = \frac{1 - c}{c} \frac{P(s = 1|\boldsymbol{x})}{1 - P(s = 1|\boldsymbol{x})}, \quad (30)$$

where $c = P(s = 1|y = 1)$. Then,

$$\mathbb{E}_{\boldsymbol{x}, y, s}[h(\boldsymbol{x}, y)] = \int_{\mathcal{X}} P(\boldsymbol{x}) \left( P(s = 1|\boldsymbol{x})h(\boldsymbol{x}, 1) + P(s = 0|\boldsymbol{x}) \Big( P(y = 1|\boldsymbol{x}, s = 0)h(\boldsymbol{x}, 1) + P(y = 0|\boldsymbol{x}, s = 0)h(\boldsymbol{x}, 0) \Big) \right).$$

The plugin estimator of $\mathbb{E}_{\boldsymbol{x}, y, s}[h(\boldsymbol{x}, y)]$ is

$$\frac{1}{n_{tr}} \left( \sum_{\boldsymbol{x}, s = 1} h(\boldsymbol{x}, 1) + \sum_{(\boldsymbol{x}, s = 0)} w(\boldsymbol{x})h(\boldsymbol{x}, 1) + (1 - w(\boldsymbol{x}))h(\boldsymbol{x}, 0) \right), \quad (31)$$

where $w(\boldsymbol{x}) = P(y = 1|\boldsymbol{x}, s = 0) = \frac{1-c}{c} \frac{P(s=1|\boldsymbol{x})}{1-P(s=1|\boldsymbol{x})}$. Thus, PU learning can be regarded as importance weighting according to the probability of labeling unlabeled data.

Owing to its theoretical underpinning, numerous PU Learning studies have embraced this weighting approach (Liu et al., 2003; Du Plessis et al., 2015; Kiryo et al., 2017; Chen et al., 2021; Du Plessis et al., 2014). Gu et al. (2022) propose Entropy Weight Allocation (EWA), which is an instance-dependent weighting method. EWA constructs an optimal transport from unlabeled instances to labeled instances and uses the transport distance to perform weighting. Acharya et al. (2022) are applying the recently popularized learning framework called contrastive learning (Tian et al., 2020; Khosla et al., 2020) in the PU learning setting.

# 7 Label Noise Correction

Label noise correction (Nicholson et al., 2016; 2015; Song et al., 2022) in machine learning involves identifying and correcting errors or inaccuracies in labeled training data for supervised learning tasks. This approach is particularly beneficial when working with datasets prone to manual labeling errors or when utilizing data from various sources that exhibit differing levels of accuracy.

Among the leading methods for label noise correction, Patrini et al. (2017) proposed a technique that uses a matrix corresponding to the probability of label noise occurrence to weight the loss function. This method can be viewed as applying importance weighting that downweights instances with a high probability of label noise occurrence. Now, assume that each label $y$ in the training set is flipped to $\tilde{y} \in \mathcal{Y}$ with probability $P(\tilde{y}|y)$, and $P(\boldsymbol{x}, \tilde{y}) = \sum_y P(\tilde{y}|y)P(y|\boldsymbol{x})P(\boldsymbol{x})$. Denote by $T \in [0,1]^{K \times K}$ the noise transition matrix specifying the probability of one label being flipped to another, where $K \in \mathbb{N}$ is the number of classes. Here, we assume that the output of a hypothesis $h$ is the vector value as $h : \mathcal{X} \to \mathbb{R}^K$, and let $\boldsymbol{\ell} : [0,1]^K \times \mathcal{Y} \to \mathbb{R}^K$ be the vector-valued loss function. Then, Patrini et al. (2017) utilizes the weighted loss as

$$\tilde{\boldsymbol{\ell}}(\mathrm{softmax}(h(\boldsymbol{x})), y) := T^{-1}\boldsymbol{\ell}(\mathrm{softmax}(h(\boldsymbol{x})), y),$$

where $\mathrm{softmax}(\cdot)$ is the softmax function. Indeed,

$$\mathbb{E}_{\tilde{y}|\boldsymbol{x}}\left[\tilde{\boldsymbol{\ell}}(\mathrm{softmax}(h(\boldsymbol{x})), y)\right] = \mathbb{E}_{y|\boldsymbol{x}}\left[T\tilde{\boldsymbol{\ell}}(\mathrm{softmax}(h(\boldsymbol{x})), y)\right] = \mathbb{E}_{y|\boldsymbol{x}}\left[TT^{-1}\boldsymbol{\ell}(\mathrm{softmax}(h(\boldsymbol{x})), y)\right]$$
$$= \mathbb{E}_{y|\boldsymbol{x}}\left[\boldsymbol{\ell}(\mathrm{softmax}(h(\boldsymbol{x})), y)\right],$$

and the corrected loss is unbiased.

Since the noisy class posterior can be estimated by using the noisy training data, the key problem is how to effectively estimate the transition matrix $T$. Given only noisy data, $T$ is unidentifiable without any knowledge (Xia et al., 2019). The key assumption of the current state-of-the-art methods is the transition matrix is class-dependent and instance-independent (Han et al., 2018b;a; Natarajan et al., 2013; Chuang et al., 2017).

Noise Attention Learning (Lu et al., 2022a) automatically models the probability of label noise using attention and employs it for weighting during gradient updates. In a similar vein, Attentive Feature Mixup (AFM) (Peng et al., 2020) combines an attention mechanism with mixup, a well-known data augmentation technique, to perform weighting based on the probability of label noise occurrence.

## 8 Subgroup Fairness

Fairness in machine learning is a research area that deals with the problem of disparity in prediction due to sensitive attributes such as race, gender, and disability (Caton & Haas, 2020; Pessach & Shmueli, 2022; Barocas et al., 2023). Interest in such fair learning is growing as various real-world applications of machine learning algorithms progress, such as medical image analysis (Mehta et al., 2024; McCradden et al., 2020), credit scoring (Kozodoi et al., 2022), or face recognition (Menezes et al., 2021). In particular, the problem of aiming for parity in prediction performance among subgroups is called subgroup fairness (or group fairness) (Kearns et al., 2019; Martinez et al., 2021; Shui et al., 2022b;a). The following is a definition of the problem setting of subgroup fairness by Lahoti et al. (2020)

**Definition 10** (Subgroup Fairness). Let $D = \{(\boldsymbol{x}_i, y_i)\}_{i=1}^{n_{tr}}$ be the dataset, and assume that there exist $K$ protected groups where each example $\boldsymbol{x}_i$, there exists an unobserved random variable $s_i$ over $\{k\}_{k=0}^K$. The set of examples with membership in group $s$ given by $D_s = \{(\boldsymbol{x}_i, y_i) \mid s_i = s\}_{i=1}^{n_{tr}}$. Then, the goal of subgroup fairness is to obtain a hypothesis $h \in \mathcal{H}$ that is fair to groups.

For the formalization of Sugroup fairness, for example, the following Rawlsian Max-Min fairness is known.

**Definition 11** (Rawlsian Max-Min Fairness (Rawls, 2001)). A hypothesis $h^* \in \mathcal{H}$ is said to satisfy Rawlsian Max-Min fairness principle if it satisfies the following.

$$h^* = \underset{h \in \mathcal{H}}{\arg\min} \max_s L_{D_s}(h), \tag{32}$$

where $L_{D_s}(h) = \mathbb{E}_{(\boldsymbol{x}, y) \sim D_s}\left[\ell(h(\boldsymbol{x}), y)\right]$.

To solve this problem, Lahoti et al. (2020) consider the following min max problem.

$$J(h, \lambda) := \min_h \max_\lambda \sum_s \lambda_s L_{D_s}(h), \tag{33}$$

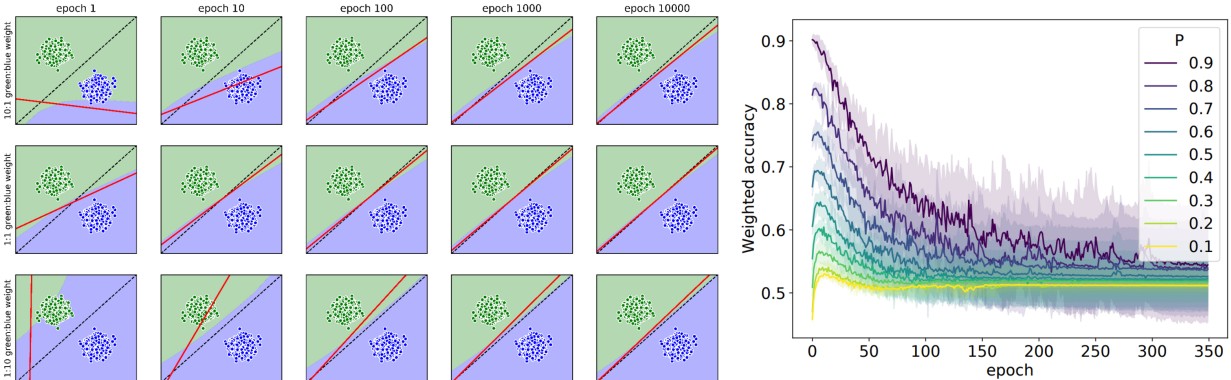

Figure 5: Left panel: decision boundaries over epochs of training. Right panel: the effect of early stopping with importance weighting. These figures are from Figure 1 and 5 of Byrd & Lipton (2019), and are reprinted with permission from the authors.

where $0 \leq \lambda_s \leq 1$ and $\sum_s \lambda_s = 1$. They propose an adversarial algorithm framework in which $\lambda_s$ is a parametric learnable function and the above is optimized to improve the worst case. This can be related to the importance weighting for each subgroup.

Also, many studies on fairness have been discussed in relation to distribution shifts such as covariate shift, and many of them explicitly utilize importance weighting (Havaldar et al., 2024; Martinez et al., 2021; Kitsuki & Managi, 2023).

## 9    Importance Weighting and Deep Learning

The behavior of Importance Weighted ERM in over-parametrized deep neural networks was unclear. Byrd & Lipton (2019) recently reported that the efficacy of importance weighting diminishes over extended iterative learning. Additionally, their experiments indicated that L2 regularization and batch normalization partially restore the effectiveness of importance weighting, but the extent of this effect depends on parameters unrelated to importance weighting. Figure 5 shows the convergence of decision boundaries over epochs of training, and the effect of early stopping with importance weighting. Their results suggest that as training progresses, the effects due to importance weighting vanish. Specifically, while importance weighting impacts a decision boundary of model early phase in training, in the limit, models with several weighting appear to approach similar solutions. They also report that these findings hold for both simple convolutional neural networks and state-of-the-art transformer-based models.

They also provide an intuitive discussion that connects the experimental results obtained with the implicit bias (Soudry et al., 2018; Ji & Telgarsky, 2018b;a; Nacson et al., 2019) of the gradient descent method. Xu et al. (2021) took this argument further, providing theoretical support for their experimental results. Through their theoretical analysis, they characterized the impact of importance weighting on the implicit bias of gradient descent and provided the generalization bound that hinges on the importance weights. Their findings suggest that, in finite-step training, the impact of importance weighting on the generalization bound is dependent on its effect on the margin and its role in balancing the source and target distributions.

Several studies have aimed at implicitly learning importance weightings using neural networks. Ren et al. (2018) introduced an online meta-learning algorithm aimed at adjusting the weights of training examples, with the goal of training deep learning models that are more robust. Their objective is

$$\boldsymbol{w}^* = \arg\min_{w} \frac{1}{n_{tr}} \sum_{i=1}^{n_{val}} \ell(h_{\boldsymbol{w}}^*(\boldsymbol{x}_i), y_i)), \tag{34}$$

where $n_{val} > 0$ is the size of validation sample, and

$$h_w^* = \arg\min_h \sum_{i=1}^{n_{tr}} w_i \ell(h(\boldsymbol{x}_i), y_i). \tag{35}$$

Through experiments, they empirically demonstrated that their proposed method outperforms existing explicit weighting techniques. For the End-to-End approach, Fang et al. (2020) reported that approximating the weighting function with a neural network induces bias. That is, given that the weight estimator $w(\boldsymbol{x})$ and weighted classifier $h_w(\boldsymbol{x})$ are each represented by neural networks, the poor performance of one negatively affects the performance of the other, since $w(\boldsymbol{x})$ and $h_w(\boldsymbol{x})$ are dependent on each other. Therefore, they proposed to address this issue by optimizing the weighting neural network and the classifier alternately as follows:

i) Define the classification model $h$ and the weight estimator $w$ as

$$h \coloneqq f \circ g,$$
$$w \coloneqq \pi \circ g,$$

where $g \colon \mathcal{X} \to \mathbb{R}^d$ is a common feature extractor, $f \colon \mathbb{E}^d \to \mathcal{Y}$ is a classification layer, and $\pi \colon \mathbb{R}^d \to [0, 1]$ is a weight estimation layer.

ii) Update the classification model with the weighted loss function $w(\boldsymbol{x})\ell(h(\boldsymbol{x}), y) = w(\boldsymbol{x})\ell(f \circ g(\boldsymbol{x}), y)$.

iii) Update the weight estimator model with fixed parameters for the feature extractor $g$.

iv) Iterate steps ii) and iii) alternately.

They show that this framework, called Dynamic Importance Weighting (DIW), can resolve the circular dependency between weighted classifiers and weight estimators.

Holtz et al. (2022) reported that learning through importance weighting improves robustness against adversarial attacks (Chakraborty et al., 2021; Xu et al., 2020; Huang et al., 2017; Guo et al., 2019). They propose a method to acquire this weighting through an adversarial learning framework.

As an alternative to importance weighting, Lu et al. (2022b) propose importance tempering to enhance the decision boundary of over-parametrized neural networks. The central idea behind importance tempering is to introduce a temperature parameter into the softmax, which corresponds to importance weighting. They reported its effectiveness in training over-parametrized neural networks.

Other research areas include importance weighting for generative models. Burda et al. (2015) pointed out that the criterion of original Variational Auto Encoder (VAE) (Kingma & Welling, 2013) is too strict, and proposed Importance Weighted VAE (IWVAE) as a variant. The evidence lower bound (ELBO) maximized by the VAE is

$$\log p(\boldsymbol{x}) \geq \mathbb{E}_{\boldsymbol{z} \sim q(\boldsymbol{z}|\boldsymbol{x})} \left[ \log \left( \frac{p(\boldsymbol{x}, \boldsymbol{z})}{q(\boldsymbol{z} \mid \boldsymbol{x})} \right) \right] = L_{\text{VAE}}(q), \tag{36}$$

and IWAE maximizes the following multi-sample ELBO.

$$\log p(\boldsymbol{x}) \geq \mathbb{E}_{\boldsymbol{z}_1, \dots, \boldsymbol{z}_k \sim q(\boldsymbol{z}|\boldsymbol{x})} \left[ \log \left( \frac{1}{k} \sum_{i=1}^{k} \frac{p(\boldsymbol{x}, \boldsymbol{z}_i)}{q(\boldsymbol{z}_i \mid \boldsymbol{x})} \right) \right] = L_{\text{IWAE}}(q). \tag{37}$$

Given a batch of samples $\boldsymbol{z}_2, \dots, \boldsymbol{z}_k$ from $q(\boldsymbol{z} \mid \boldsymbol{x})$, we can see that IWVAE utilizes the following importance weighted distribution.

$$\tilde{q}(\boldsymbol{z} \mid \boldsymbol{x}, \boldsymbol{z}_{2:k}) = \frac{\frac{p(\boldsymbol{x}, \boldsymbol{z})}{q(\boldsymbol{z}|\boldsymbol{x})}}{\frac{1}{k} \sum_{j=1}^{k} \frac{p(\boldsymbol{x}, \boldsymbol{z}_j)}{q(\boldsymbol{z}_j|\boldsymbol{x})}} q(\boldsymbol{z} \mid \boldsymbol{x}). \tag{38}$$

Note that when $k = 1$, $\tilde{q}$ is equal to $q$, and as $k \to \infty$, $\mathbb{E}_{z_2, \ldots, z_k}[\tilde{q}(\boldsymbol{z} \mid \boldsymbol{x}, \boldsymbol{z}_{2:k})]$ approaces the true posterior $p(\boldsymbol{z} \mid \boldsymbol{x})$ pointwise. IWAE has been the subject of much discussion due to its effectiveness. Rainforth et al. (2017) and Nowozin (2018) provide the analysis of IWVAE bounds and other Monte Carlo objectives. Daudel et al. (2023) generalized IWVAE by $\alpha$-divergence and characterized its statistical properties.

## 10 Conclusion and perspective

In this review, we explored the application of importance weighting operations within the realm of machine learning. Notably, certain methodologies not traditionally identified as importance weighting can be reinterpreted as such operations (for instance, focal loss in model calibration and label noise correction). Moreover, numerous studies, particularly in recent times, have introduced akin methods without directly referencing Importance Weighted Empirical Risk Minimization (IWERM) (Shimodaira, 2000). Viewing these diverse techniques through the lens of importance weighting offers several benefits.

  i) **Utilization of existing results**: Many theoretical results are known for IWERM. For example, Shimodaira (2000) gives the information criterion for IWERM, and Cortes et al. (2010) provides a generalization error analysis of importance weighted supervised learning. By interpreting one technique as importance weighting, we can take advantage of these existing theoretical results.

 ii) **Determination of weighting functions**: When a technique can be considered as importance weighting, the key question is how to determine the weighting function. As we observed in the examples of importance sampling in Section 1 and unbiasedness of IWERM under the covariate shift in Section 3.1, the selection of the weighting function is strongly linked to the density ratio. When the optimal weighting function for a given method can be expressed using density ratio, we may be able to take advantage of density ratio estimation techniques as reviewed in Section 2.1.

iii) **Sharing negative results**: Several studies have reported that importance weighting does not work well in some cases (e.g., Byrd & Lipton (2019)). We can take these negative results into account when considering the experimental results of methodologies that can be considered as importance weighted learning.

While there are many advantages to a unified interpretation as a framework of importance weighting as described above, it is possible that many of the methods not covered in this survey could also be considered as IWERM and its variants. For example, the IWAN (Zhang et al., 2018b) in Section 3.6.2 can be shown to reproduce the relative importance weighting (Yamada et al., 2013) by a simple calculation, but this fact is not mentioned in the original IWAN paper. However, the relative importance weighting has been shown to be a density ratio approximation based on minimizing Pearson divergence, and several theoretical results are known. This suggests that some techniques can be taken a step further by leveraging existing known results, and unified interpretation through the importance weighting framework makes this possible.

The error bound is an important topic to discuss. For example, DIW (Fang et al., 2020), one of the experimentally successful importance weighting methods in deep learning, uses Kernel Mean Matching (KMM) for updating the weight estimator, and its error bound is $\mathcal{O}(n_{tr}^{-\frac{r}{2r+4}} + n_{te}^{-\frac{r}{2r+4}})$. Li et al. (2020a) propose a robust importance weighting method for covariate shift, and achieves $\mathcal{O}(n_{tr}^{-\frac{r}{2r+2}} + n_{te}^{-\frac{r}{2r+2}})$, which is superior to the best-known rate of KMM. However, currently only a limited number of studies provide the error bounds of each method.

Another future work would be to analyze the behavior of importance weighting in more modern machine learning problem settings. In Section 9, we mentioned studies on the behavior of importance weighting in deep learning. However, the behavior of importance weighting in large neural networks, such as the Large Language Models (LLM) (Brown et al., 2020; Floridi & Chiriatti, 2020; Chang et al., 2023) that have recently achieved significant results, is not yet understood. Theoretical and experimental verification of which cases of importance weighting work well and which fail would be very useful. As we mentioned, while many of the techniques can be considered special variants of importance weighting, they do not seem to make full use of the known results. We hope that this survey is useful to address these unresolved issues.

## Acknowledgement

Part of this work is supported by JST CREST (JPMJCR2015), JST Mirai Project (JPMJMI21G2), and JSPS KAKENHI(JP22H03653). Finally, we express our special thanks to the editor and anonymous reviewers whose valuable comments helped to improve the manuscript.

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

## A   Proof of Lemma 1

*Proof.* From the assumption, we have $P(s = 1|y = 1, \boldsymbol{x}) = P(s = 1|y = 1)$. We also have

$$
\begin{aligned}
P(s = 1|\boldsymbol{x}) &= P(y = 1 \wedge s = 1|\boldsymbol{x}) \\
&= P(y = 1|\boldsymbol{x})P(s = 1|y = 1, \boldsymbol{x}) \\
&= P(y = 1|\boldsymbol{x})P(s = 1|y = 1).
\end{aligned}
\tag{39}
$$

By dividing each side by $P(s = 1|y = 1)$, we can obtain the proof. $\qquad\square$

