# OpenReview forum: "A Short Survey on Importance Weighting for Machine Learning"
_TMLR — Accepted by TMLR_

### Review · Reviewer_mPi6 · 2024-03-27

**Summary Of Contributions:**

This paper provides an overview of the concept of importance weighting, its applications, and related research in machine learning. The paper discusses various methods for handling distribution shift, such as covariate shift, target shift, and sample selection bias, highlighting the effectiveness of importance weighting in these scenarios. The authors also present the role of importance weighting in addressing various machine learning tasks such as active learning, model calibration and so on.

**Audience:**

Yes

**Broader Impact Concerns:**

N/A.

**Claims And Evidence:**

Yes

**Requested Changes:**

Please refer to Weaknesses.

**Strengths And Weaknesses:**

Strengths:

1. This survey covers a broad range of topics related to importance weighting, making it a valuable resource for researchers and practitioners.

2. The authors provide a clear and accessible explanation of complex concepts, making the material understandable to readers with varying levels of expertise. The inclusion of both theoretical and practical aspects of importance weighting offers a good view.

Weaknesses:

1. While the survey is comprehensive, it may benefit from more detailed case studies or examples to illustrate the practical applications of importance weighting in real-world scenarios. This survey covers a broad range of importance weighting methods but none of these has been formally introduced in details.

2. There are several confused points about the paper structure: (a) Why Density Ratio Estimation is missed from the structural diagram in Figure 1? (b) Domain Adaptation, which is inherently a specific type of Distribution Shift Adaptation, is presented as a separate entity in the paper. (c) It seems that Distributionally Robust Optimization is one part of Distribution Shift Adaptation within the diagram (Figure 1), but the Section of Distributionally Robust Optimization is positioned between Active Learning and Model Calibration sections.

3. The paper could potentially include a more extensive discussion on the computational complexity and scalability of the various importance weighting methods, which are crucial considerations for large-scale machine learning tasks.

4. Although the survey mentions the use of importance weighting in deep learning, it may not fully capture the details of its utilization in deep learning. I think this part could be extend considering that deep learning is dominating modern machine learning.

---

> ### Author Response · Authors · 2024-04-16
> **Authors' Response**
>
> Let us thank you for your very informative comments. You have accurately grasped the content of our manuscript and have given us enough comments to improve our manuscript. The following is a list of our revisions in response to the points you have raised.
>
> 1. Illustration of practical applications and formal definitions of them.
> - To address these points, we first created and added new figures to provide a conceptual understanding of the problem settings for which importance weighting has been reported to be effective (see Figure 2 in p.6 and Figure 3 in p.12). In these figures, we have tried to show the relationships and connections between each problem setting, rather than presenting each problem setting independently. We believe these figures will help to understand what kinds of problem settings the importance weighting solves. Additionally, in response to your point about the lack of formal details, we have added formal definitions of the various problem settings (see Definition 6 in p.8, Definition 7 in p.9, Definition 8 in p.10, Definition 9 in p.11, Definition 10 in p.16 and Definition 11 in p.17). In these definitions, we have kept the symbols consistent with the other definitions, but have tried to distinguish them from other similar concepts. As above, we have attempted to help the reader by not only enumerating the concepts, but also by making explicit the connections between each of them.
> 2. Confusing structure:
> - a) Why Density Ratio Estimation is missed from the structural diagram in Figure 1?
>   - While Figure 1 lists the applications of importance weighting, density ratio estimation was not originally included in Figure 1 because it is a technique that supports importance weighting. However, the original manuscript was confusing because the section on density ratio estimation was presented in parallel with the other sections. In light of this, we have moved this section to a subsection of the preliminary section in order to clarify the position of density ratio estimation as a fundamental technique for importance weighting in our revised version (see Section 2.1). We have also clarified the meaning of this subsection by writing at the end of this subsection that many importance-weighting-dependent techniques depend on density ratios.
> - b) Domain Adaptation, which is inherently a specific type of Distribution Shift Adaptation, is presented as a separate entity in the paper.
>   - As you pointed out, it is reasonable to position domain adaptation as a sub-problem of distribution shift. Therefore, we have moved the Domain Adaptation section to a subsection of the Distribution Shift section (see 3.6). We have also clarified the distinction between other distribution shifts and domain adaptation assumptions by specifying the difference between them immediately below Definition 8 (domain adaptation) on p. 10.
> - c)  It seems that Distributionally Robust Optimization is one part of Distribution Shift Adaptation within the diagram (Figure 1), but the Section of Distributionally Robust Optimization is positioned between Active Learning and Model Calibration sections.
>   - As you pointed out, we have moved the Distributionally Robust Optimization section to a subsection of the Distributional Shifts section (see Section 3.7). At this point, we made sure that the order of the topics appearing in Figure 1 matches the order of the sections.
> 3. Discussion on the computational complexity.
> - Computational complexity is a very important issue, as you suggest, and we found it worth devoting a paragraph to in the Conclusion section. In particular, we mentioned an example of improved computational complexity for importance weighting and the need for such an analysis in future works.
> 4. More discussion on importance weighting and deep learning
> - Your comments on the usefulness of the section on importance weighting and deep learning are reasonable. We first tried to enhance the discussion on one of the most important topics in this section: the negative results of importance weighting. The point of this topic is that the effect of importance weighting is decaying in learning by gradient descent over long periods of time. This observation is so illuminating that we have decided to devote more space to it. Therefore, we contacted the authors of the original paper that reported these results and obtained permission to use the figures from their paper (see Figure 5). Next, we introduce the existing studies in more detail on learning weight estimators, which is an important issue for importance weighting in deep learning. Finally, we added a new paragraph on Importance Weighted VAE, which has many related studies and touches on the topics of importance weighting and generative models.

---

> > ### Comment · Reviewer_mPi6 · 2024-05-03
> >
> > Thank you for your response.
> >
> > I am pleased that you have made modifications to the organizational structure, figures, and other content in the revised version. I believe that the revised paper is clearer and more readable than the previous version.

---

### Review · Reviewer_7PJm · 2024-03-30

**Summary Of Contributions:**

This short survey paper discusses the importance weighting in machine learning, a very popular and practical tool in the context of distribution shift. This paper systematically conducted the analysis on importance weighting in different aspects.

**Audience:**

Yes

**Broader Impact Concerns:**

N/A.

**Claims And Evidence:**

Yes

**Requested Changes:**

### Page 2
- In Fig 1. Each term in applications of IW could be associated with the concrete sections (e.g., Target shift (Sec 3.2)).
- In Sec 2. The definitions 1 and 2 could be simplified. Indeed, this is well-known for most readers in ML, it could be simplified.

### Page 3
- As for the distribution shift adaptation. I think this title seems a bit confusing because Sec 4 is also called domain adaptation. Is there any difference between the two adaptations?
- In Sec 3.1 (the proof can be put into appendix). Authors are encouraged to include some figures in the covariate shift.
- In IWERM, RIERN, AIWERM. It is much better to make a table to compare and discuss.

### Page 4
- The derivation of Eq(11) can be put in the appendix.
- The Dro paragraph could be merged with the DRO later.
- It is better to show some figures to illustrate the target shift. (This could be discussed together with covariate shift).

### Page 5
- Sec 3.3 The sample selection bias has no formal definition.
- It is better to illustrate how to use this in machine learning.
- Sec 3.4. No definitions for subpopulation (or subgroup) shift.
- There is a notation error on the bottom of page 5. What is x_i,j. It is a typo?

### Page 6
- The definition of x_i, x_j, y_i and y_j are finally defined. But it should be present at Page 5.
- Sec 3.5. It is much better to provide a formal definition of feedback shit. Please note that these differences shifts are related. It is important to clarify and discuss their differences.
- Sec 4. There is no definition on domain adaptation.

### Page 7
- In sec 4.1, the multi-source domain adaptation could have different views on IW. It can be the weights on the domains or weights on the covariates/targets. For example, paper [1] discussed two weights.
- In sec 4.2, partial DA. I would think paper [1] also bridged the IW with partial DA. Indeed, partial DA could be viewed as a specific target shift problem. If we have a weight coefficient, it is just some coefficient that has zero elements, due to the partial labeling.
- Bottom of Page 7, a typo ((Yanma et.al 2013)) -> (Yanma et.al 2013).

### Page 8
- Sec 4.3, 4.4. It could be better to use a formal definition to discuss different DA settings.

### Page 9
- In sec 5. The AL can be viewed as IW in an alternative way. For example, if we design a query distribution $q(x)$ (something we want to learn) to match the entire unlabelled data $D(x)$.(e.g, see paper [2]). If we set a distribution divergence metric as Renyi (i.e, min_q Renyi(q(x), D(x))), I think this could be recovered to some IW methods.

### Page 10
- In sec 10, model calibration. It is strongly recommended to use some figures and tables to illustrate calibration and focal loss, adaptive focal loss.

### Page 11
- Proof of Lemma 1 can be put in the appendix.
- Eq(22) is the demonstration rather than the derivation, right?


### Page 12
- Sec 10. I think this part needs major revisions. I know all the material but I could not get the point of IW. Or I do not know how to use this in machine learning…

### Page 13
- Sec 11. It can be a bit more detailed about this important issue.

### Page 14
- The conclusion seems a bit short. I would like to see clear future directions on IW.
- Besides, some topics related to IW seem missing. For example, (1) IWVAE, using importance weighting in the generative model such as paper [3] . (2) Fairness and subgroup discriminations such as paper [4,5,6].


References:

[1] Aggregating From Multiple Target-Shifted Sources. ICML 2021

[2] Deep Active Learning: Unified and Principled Method for Query and Training. Aistats 2020.

[3] Importance Weighted Autoencoders. ICLR 2015

[4] Fair Representation Learning through Implicit Path Alignment. ICML 2022

[5] On Learning Fairness and Accuracy on Multiple Subgroups. NeurIPS 2022

[6] Fairness without demographics through adversarially reweighted learning. NeurIPS 2020.

**Strengths And Weaknesses:**

### Strong point

First of all, I really like this short survey. This paper is a timely summary of Importance Weighting (IW) in various aspects. I am familiar with some concepts and others are quite new to me. The writing is quite accessible to me.

### Weak point

Since this is a survey paper, there are no technical concerns. However, such a paper generally has a very high standard of organization and illustration. Please check the requested changes as the concerns during my reading. I hope this could be helpful to improve this survey paper.
**Please note that all the questions in the request changes are related to the clarifications. In fact, I fully understand the concept, it is more on the issue of clarity.**

---

> ### Author Response · Authors · 2024-04-16
> **Author's Response 1/2**
>
> We are very pleased with your positive and informative comments. In particular, all of the topics you have suggested are very interesting and we are very excited to be able to improve the manuscript by adding them. As you have specified the suggested corrections on a page-by-page basis, we list below the corresponding corrections accordingly. Please note that the original page numbers and the page numbers of the revised version have been shifted by the additions and corrections.
> ## Page 2
> - In Figure 1, each topic is now associated with a section number.
> ## Page 3
> - We moved the Domain Adaptation section as a subsection of Distribution Shift Adaptation. We have also clarified the distinction between other distribution shifts and domain adaptation assumptions by specifying the difference between them immediately below Definition 8 (domain adaptation) on p. 10.
> - We have created and added a figure so that the two distribution shifts, covariate shift and target shift, can be understood conceptually (see Figure 2).
> - We have created a table to provide a summary of the properties of IWERM, AIWERM, and RIWERM.
> ## Page 4
> - We merged DRO paragraph with the later section.
> - In Figure 2, the target shift is illustrated together with the covariate shift.
> ## Page 5
> - We provided the formal definition of sample selection bias (see Definition 5).
> - We added Theorem 1 to illustrate the usefulness of importance weighting under sample selection bias. This theorem shows that correction of sample selection bias can be achieved with appropriate importance weighting.
> - We provided the formal definition of subpopulation shift (see Definition 6).
> - We fixed a typo.
> ## Page 6
> - We moved the definition of x_i, x_j, y_i and y_j to a more favorable position.
> - We provided the formal definition of feedback shift (see Definition 7).
> - We provided the formal definition of domain adaptation. We also clearly stated the difference between domain adaptation and other distribution shifts.
> ## Page 7
> - We have added the discussion of the multi-source domain adaptaiton as you suggested.
> - As you pointed out, it seems that partial domain adaptation can be related to multi-source domain adaptation. We have therefore added to this point, citing the references you suggested.
> - We fixed a typo.
> ## Page 8
> - In order to clarify the differences between the problem settings for each domain adaptation, we have created a new figure (see Figure 3). This figure provides a conceptual understanding of what assumptions are made for each domain adaptation subproblem.
> ## Page 9
> - Reading the literature you suggested and your comments, we found it useful to relate a different perspective on active learning and importance weighting. In fact, the discussion of active learning and importance weighting from this perspective is so intriguing to us and we have added it by citing the literature that has suggested.
> ## Page 10
> - We have created a new figure on the motivation for model calibration and the behavior of focal loss (see Figure 4). The left panel of this figure shows what a well-calibrated model looks like and identifies the goals of model calibration. The right panel of this figure shows the difference in behavior as the focal loss parameter is varied, and for certain values of the parameter, the behavior is consistent with cross entropy.
> ## Page 11
> - We moved the proof of Lemma 1 in the appendix.
> - As you pointed out, Eq(22) is the demonstration rather than the derivation. We have clearly stated this in the revised version.

---

> > ### Author Response · Authors · 2024-04-16
> > **Authors' Response 2/2**
> >
> > ## Page 12
> > - As you pointed out, the section on density ratio estimation did not connect well with the other sections in the original manuscript. The main reason for this is that density ratio estimation is the supporting technique for importance weighting while the other sections focus on applications of importance weighting. Although density ratio estimation is an inseparable topic from importance weighting, it was confusing to discuss it in parallel with the other sections. Therefore, we have moved the section on density ratio estimation to a subsection of the preliminaries section and specified its meaning (see Section 2.1).
> > ## Page 13
> > - As you suggested, the section on importance weighting and deep learning should have been given more space for discussion. The point of this topic is that the effect of importance weighting is decaying in learning by gradient descent over long periods of time. This observation is so illuminating that we have decided to devote more space to it. Therefore, we contacted the authors of the original paper that reported these results and obtained permission to use the figures from their paper (see Figure 5). Next, we introduce the existing studies in more detail on learning weight estimators, which is an important issue for importance weighting in deep learning. Finally, we added a new paragraph on Importance Weighted VAE which you suggested to add.
> > ## Page 14
> > - We have added a paragraph in the conclusion section on the important topics of importance weighting and computational complexity. We believe that presenting such a topic on complexity is of great meaning in the context of the recent trend toward larger data and models. Furthermore, we emphasized that many existing studies have not been able to use the existing results, even though they can be regarded as importance weighting procedures.
> > - A couple of topics regarding importance weighting:
> > - 1) IWVAE: We added this topinc in Section 9.
> > - 2) Subgroup fairness: We found this topic you introduced so interesting that we decided to devote a section to it (see Section 8). In particular, we found similarities between the approach used in fairness and the importance weighting framework in other problem settings, and we feel that this is a topic that should have been introduced in this manuscript.

---

> > > ### Comment · Reviewer_7PJm · 2024-04-16
> > >
> > > Thank you! I quickly checked the revised the paper and I feel the new version seems much better. Most concerns are properly addressed.
> > >
> > > I have a follow-up remark on Fig 2. Indeed this figure seems still a bit confusing. As far as I understand, in the target shift, it should be the ground-truth label distribution shift ($Y$ shift). In figure 2, it seems providing a wrong impression such that the predicted label distribution shift ($\hat{Y}$ shift). A clarification on this could be better.

---

> > > > ### Author Response · Authors · 2024-04-17
> > > > **Authors' Response**
> > > >
> > > > Many thanks for checking the revised version of the manuscript quickly and for the additional suggestions.
> > > > As you mentioned, the explanation for the target shift in the original Figure 2 was confusing. Therefore, in the newly recreated figure we have clarified that the target shift is a shift in the ground truth distribution, and then added text to make it explicit in the figure. Thanks to your comments, we believe the concepts related to the distribution shift are much clearer now.

---

### Review · Reviewer_3jUS · 2024-04-04

**Summary Of Contributions:**

This paper aims to review certain machine learning problems related to sample weighting. By presenting a previous definition and mathematical model of importance weighting, the paper reviews many pieces of machine learning problems using weighting to tackle challenges, thereby transforming these problems into a unified weighting technique.

**Audience:**

Yes

**Broader Impact Concerns:**

No.

**Claims And Evidence:**

Yes

**Requested Changes:**

I believe there is ample opportunity for improvement in the survey by not merely introducing the topics and concluding.

**Strengths And Weaknesses:**

As a survey paper, this paper demonstrates a solid formulation and its linkage to various machine learning problems. However, as a survey, I believe there should be more analysis of the different works rather than just listing them in a table. For instance, readers might be interested in understanding which issues are effectively addressed by importance weighting and what challenges still remain for each machine learning problem utilizing importance weighting. Additionally, a survey should provide a comprehensive overview of all the problems and draw insightful conclusions regarding future directions or challenges within the surveyed topic.

---

> ### Author Response · Authors · 2024-04-16
> **Authors' Response**
>
> Let us thank you for your informative comments.
> As you mentioned, the important thing for a good survey paper is to connect the existing studies well.
> Based on this, we have attempted to go beyond merely listing the existing studies and to connect them well by the following modifications.
> - Additional figures for clarification of the problem settings.
>   - We have added several figures to clarify the similarities and differences between the multiple problem settings that the importance weighting addresses (Figure 2 in p.6 and Figure 3 in p.12). First, Figure 2 shows two different distribution shifts: covariate shift and target shift. These differences can be expressed by placing the shift assumption on the input or output variable, and we have attempted to create a figure that allows conceptual understanding of this. Next, Figure 3 provides an overview of the problem settings for different domain adaptations. In particular, these domain adaptation settings can be confusing, and we believe that organizing them in a single figure will help us to understand the connections among the existing studies.
> - Table for comparison of methods.
>   - We created a table to compare IWERM, AIWERM, and RIWERM, which are the most important groups in the importance weighting framework (Table 1 in p.5). This table specifies the advantages and disadvantages of each method.
> - Addition of formal definitions.
>   - We have added formal definitions to clarify the position of each problem setting (see Definition 6 in p.8, Definition 7 in p.9, Definition 8 in p.10, Definition 9 in p.11, Definition 10 in p.16 and Definition 11 in p.17). We have tried to introduce each of these definitions in a way that is easy to understand what is similar and what is different.
> - Addition to the conclusion section.
>   - In the conclusion section, we highlighted that although many methods can be applied in a importance weighting framework, they do not exploit the existing results. Indeed, as we demonstrated in Eq. 24 in Section 3.6.2 in p.11, adversarial learning-based domain adaptation can be associated with RIWERM, and PU learning can be considered to follow an importance-weighting framework, as shown in Eq.31 in Section 6 in p.15. In this way, we demonstrated that existing methods can actually be rewritten as a importance weighting procedure, and emphasized the usefulness of bringing existing methods together from a unified perspective in this manner.

---

### Author Response · Authors · 2024-04-16
**General Comments for All Reviewers**

First of all, let us express our sincere thanks to the editor and all the reviewers for taking the time to review our manuscript.
Especially to the reviewers, we have no doubt that everyone's comments are very helpful in improving our manuscripts.
Some of the comments from the three reviewers were from similar and some were from different perspectives. In our revised version, we have made modifications in response to everyone's comments.

Note:
- $\textcolor{red}{\text{All modifications are shown in red}}$.
- Figure 5 in the revised version is a combination of figures borrowed from another paper. We contacted the authors of the paper directly to obtain permission to use these figures.
- All figures except Figure 5 are created by ourselves.

We hope that our revised version is better than the original one, taking into account the points you have made.

---

### Decision · Action_Editor_BDZJ · 2024-05-09

**Recommendation:** Accept as is

**Comment:**

As the authors emphasized in the paper, importance weighting is an essential technique in machine learning.
This paper successfully summarizes the broad yet relevant topics in importance weighting.
Moreover, as the authors noted through the paper, some popular methods can be interpreted as a special case of importance weighting even though they do not explicitly state in that way.
This paper provides a unified view of these existing techniques through the lens of importance weighting, which provides several benefits listed in conclusion.

In the reviews, all the reviewers were positive towards this paper.
The authors addressed the concenrs raised by the reviewers, which effectively improved the contents of the paper.

Lastly, I would like to ask the authors to proofread the paper for the final submission.
Few corrections will be needed including typos, mostly in the updated parts, as below.

> p.3

`Kernel mean matching Huang et al. (2006); Gretton et al. (2009)` will be something like `Kernel mean matching (Huang et al. 2006; Gretton et al. 2009)`
The same goes for `(KLIEP) Nguyen et al. (2010); Sugiyama et al. (2007b)` and `(LSIF) Kanamori et al. (2009)`

> p.7, Theorem 1

If I understand correctly, this should be

$$
\mathbb{E}\_{tr}[w(x)\ell(h(x), y) | s = 1] = \mathbb{E}\_{te}[\ell(h(x), y)],
$$

which states that the IWERM under the selection bias in the left is equivalent to the test error in the right.

> p.8, Eq.(22)

This will be $\cap$ instead of $\cup$.

> p.11

A space is missing after (2018b) in `Zhang et al. (2018b)suggested`

> p.15

There is a space in the beginning of the sentence `where $w(x) = ...$` just after Eq.(31).

> p.18

``where $n_{v}al$`` will be ``where $n_{val}$``

There is a space in the beginning of the sentence `Through experiments, they empirically ...` just after Eq.(35).

> p.20

The term ``computatinal complexity`` here seems to refer something like `error bound`.
Please consider whether the use of this term is appropriate.
The term ``computatinal complexity`` generally refers to time or space complexity needed to execute an algorithm, but not the quality of the output of an algorithm such as the decay of an estimation error with respect to the data size.

**Audience:**

Yes.
The topic is relevant to TMLR because importance weighting is an essential technique in machine learning.

**Claims And Evidence:**

Yes.
As a survery paper, this paper covers broad topics of importance weighting in machine learning.
Moreover, as the authors noted through the paper, some popular methods can be interpreted as a special case of importance weighting even though they do not explicitly state in that way.
This paper provides a unified view of these existing techniques through the lens of importance weighting, which provides several benefits listed in conclusion.

---

> ### Author Response · Authors · 2024-05-14
> **Thanks to the editor and the reviewers**
>
> We would like to thank the action editor and the reviewers for taking the time to provide useful comments.